# Saddar Bazar Quarter in Karachi: A Case of British-Era Protected Heritage Based on the Literature Review and Fieldwork

**Syed Hamid Akbar** *, **Naveed Iqbal and Koenraad Van Cleempoel**

Faculty of Architecture & Arts, Hasselt University, 3500 Hasselt, Belgium
* Correspondence: syedhamid.akbar@uhasselt.be; Tel.: +32-486-393243

**Abstract:** On the international level, heritage is considered an essential element for the sustainable development of a country. In South Asian countries such as Pakistan, India, and Bangladesh, historic cities struggle to preserve their built heritage, due to rapid urbanisation and changing contemporary urban and architectural requirements. This contribution elaborates on the effects of urbanisation, and city development on the protected heritage of Karachi, Pakistan. The city developed in the 19th century from a mud-fortified small town along the Lyari river to become one of the busiest ports of the Indian subcontinent under British rule. Karachi has now become a metropolitan city with more than sixteen million inhabitants. Due to the accelerated rate of urbanisation and trade activities, Karachi has become Pakistan's economic capital, resulting in the oblivion of its historical sites. Most of the city's historic sites are in a state of dereliction, from over- or under-programming, or even under the threat of demolition. The paper focuses on the present condition of a British-era protected-heritage site through a literature review and fieldwork (mapping, drawing, archival research, interviews, photographic surveys, etc.), carried out during 2019 and 2020. The first author conducted site visits to photograph buildings and interview their users to gather feedback on how they perceive the current state of these buildings. The data were analysed to investigate how many buildings from Karachi's British era with protected status have been demolished or are at risk of demolition. At the heart of the article is the Saddar Bazar Quarter in Karachi and its historical, social, cultural, and economic importance in the city from the British period until the present. The analysis will lead the discussion on what potential these sites/buildings hold, and how to make their preservation possible and withstand the uncurbed urbanisation and the threat of land development. Through discussion, we will focus on the social, cultural and economic aspects which the area and the buildings from the British period hold, and which can be useful in the future for the continuation of the Saddar Bazaar Quarter (SBQ), the historic urban landscape (HUL) and the heritage buildings.

**Keywords:** British heritage; Saddar Bazaar Quarter; historic urban landscape; Karachi; Pakistan

## 1. Introduction

Historic buildings, which were once an integral part of the area's development, have cultural and socio-economic values from the past and can become facilitators for sustainable development in the future. However, with the growing speed and intensity of urbanisation processes, cities with historic urban fabric and sites are under the threat of ignorance, underutilisation or over-programming, and demolition, in the worst-case scenarios [1,2].

The urbanisation process is booming at an unprecedented speed, and people are moving towards big cities for better facilities. According to a United Nations report, 56% of the world population in 2022 were living in urban areas, and by the year 2050, the percentage will rise to 68% [3,4]. In South Asian countries, including Pakistan, population growth and rapid economic development are two significant factors that encourage urbanisation [5,6].

They have set the stage for pressuring rapid urbanisation and changing the city's contemporary urban and architectural requirements, resulting in the decline of the built heritage or even the destruction of the historic urban fabric [2,7,8].

While the Western world has established the conservation and adaptation of historic urban fabric and sites as standard practice, developing countries in Asia continue to debate their preservation or demolition [1,8,9]. As a developing country in South Asia, urbanisation is not a new concept in Pakistan. Since its independence, due to population growth, immigration, and economic opportunities, the country has shown rapid growth in the urban population in big cities such as Karachi, Lahore, and Faisalabad. According to UN-Habitat, Pakistan is one of South Asia's most urbanised countries, with an annual urban-population growth rate of 2.53 per cent. In addition, around 54 per cent of the urban population live in the ten major cities of Pakistan. Among them, the city of Karachi contains the lion's share [10,11]. Karachi, once a tiny village with a population of 10,000 people at the beginning of the 19th century, has now become a megacity, with a population of more than sixteen million [12,13].

From 1839 to 1947, the British ruled the city of Karachi, designing the city's urban layout and developing a unique colonial architecture. Karachi became the capital of the Sindh province and an important port for trade and economic activities during the British period. The British developed the new area in Karachi beyond the old mud town of the city on the European development pattern, but also re-developed the old area for the locals. Among the areas developed by the British, Saddar Bazar Quarter was the central hub of social, cultural and economic activities [14,15].

Along with Karachi's urban extension and development, they also introduced the European architectural style in the city through buildings such as Frere Hall. In the second half of the 19th century, this European style was combined with regional styles and influence from Mughal architecture to create a distinctive 'colonial architecture' that is still present in many government, commercial, residential, and other buildings today. In this context, the aims of the paper are:

- To investigate the development of the city's urban layout and architecture during the British era from 1839 to 1947;
- To explore how the city's historic architecture has changed and endured through various transitional stages as a result of shifting contemporary needs, from 1947 to the present;
- To present the present condition of Karachi's buildings from the British era with protected heritage status and try to find the aspects responsible for the current condition of these heritage buildings.

The focus of the study is specifically on the historic Saddar Bazar Quarter, which is selected based on the importance of the area since 1839, as the first quarter which the British established after the city's occupation in that year. Since then, the Saddar Bazaar Quarter has played a crucial role as a central hub of social, cultural and economic activities during the British period and after independence, in the development of Karachi.

### 1.1. Methodology

As the first step, a literature survey was carried out about the history of Karachi city, its origin, rulers of the area, population, and urban and architectural development, based on the three sub-sections of the study. The Section 1.1 on the pre-history of Karachi before 1839 (when the British took over the city from the Talpur rulers) helps us to understand the motivation behind the British occupation of a small mud-fortified town. The Section 2 helped understand the development and evolution of Karachi's urban layout and architectural styles under British rule from 1839 until independence in 1947, and the last sub-section of the literature survey helped to understand the post-independence development of the city and its British-era historic core and buildings, according to changing urban and user requirements until the present day. For this literature survey, different books, academic-journal articles, newspaper articles, conference-proceeding papers, PhD

theses and official gazetteers on the city of Karachi were reviewed in reference to its urban features and architecture [12,14–27].

After the literature review, initial inventories of 1995–1997, were studied to determine what buildings were given protected status, focusing again on Saddar Bazar Quarter. The inventory of the Saddar Bazar Quarter and some reports were reviewed, to see the state of heritage buildings after being given protected status [28,29].

This literature survey and collected data from different archives laid the basis for the first fieldwork in January 2019. The purpose was to visit various departments to meet people and have a general discussion. The people met were informally interviewed with open-ended questions on the present condition, heritage status and heritage management of the British-period buildings (Table 1). These interviews helped to understand the heritage legislation and its implementation system in Karachi. During the 2019 visit, an initial photographic survey was also carried out of different quarters of Karachi, focusing mainly on famous monumental buildings: Frere Hall (1865), Trinity Church (1855), St. Patrick's Cathedral (1881), Merewether Memorial Tower (1892), and Flagstaff House (1890)—now Quaid e Azam House Museum—and Mohatta Palace (1927), to see the present condition of British buildings. The Saddar Bazaar Quarter (SBQ) was among the quarters surveyed for photographing two monumental buildings, Edulji Dinshaw Dispensary (1882) and Empress Market (1889), in the area.

**Table 1.** List of Interviewees. (The interviews were conducted by Syed Hamid Akbar (first author) during his research visits to Pakistan in 2019).

| No. | Profession | Date of Interview |
|:---:|:---:|:---:|
| 1 | Architect, heritage conservationist | 29 January 2019 |
| 2 | Architect, social activist | 9 January 2019 |
| 3 | Architect, Professor at NED Karachi University | 8 January 2019 |
| 4 | Architect, Professor at COMSATS Lahore | 6 February 2019 |
| 5 | Architect, professor at SUET Karachi | 7 January 2019 |
| 6 | Architect, ICOMOS Pakistan, President | 3 February 2019 |
| 7 | Architect, ICOMOS Pakistan, Vice President | 8 January 2019 |

The second research field visit was carried out from December 2020 to January 2021. That visit aimed to conduct an in-depth study of the Saddar Bazar Quarter and its protected heritage buildings. The literature survey and the first research visit helped select 76 buildings with protected heritage status from 1995–1997 under 'The Sindh Cultural Heritage Preservation Law of 1994' [19]. The second research visit focused on collecting data by visiting different archival departments to find historic urban and architectural maps, written documents, and historical photographs (Table 2).

**Table 2.** Different Archives were visited by Syed Hamid Akbar (first author) to collect archival data, from 2019 to 2022.

| S. NO | Archive Name | Date | Data Found |
|:---:|:---:|:---:|:---:|
| 01. | Heritage Cell-DAPNED, Karachi | January 2019, December 2020 | Journal Articles, books, reports. |
| 02. | Sindh Archives, Karachi | December 2020, January 2021 | 1874 Saddar Bazaar Quarter survey sheets, old maps of Karachi. |
| 03. | Karachi Metropolitan Corporation Archives, Karachi | December 2020, January 2021 | Building records of 19th and 20th century (British period), written documents. |
| 04. | Karachi Port Trust Archives | December 2020, January 2021 | Old maps of Karachi port from the British period. |
| 05. | The British Library, London, UK. | March 2022 | Old photographs of buildings, 19th century maps of Karachi, post-independence maps of Karachi, written documents, and newspapers from the British period. |

During this visit, a photographic survey of the Saddar Bazaar Quarter was done, in which 76 buildings from the 1995–1997 inventory list were visited, and the exterior and interior of the buildings were documented; (in some cases, the building users were not allowed to take pictures from inside, so their privacy was respected). These pictures were also compared with the previous reports and the historic photos from different archives. The informal interviews were also conducted with the users and owners of the remaining and in-use buildings from the 76 buildings that were given protected heritage status from 1995–1997 under the Sindh Cultural Heritage Preservation Act of 1994. For informally interviewing building users, for each building, we tried to interview at least three people. The questions asked were semi-structured and open-ended, based on the literature review conducted for the ongoing Ph.D. research by the first author, to obtain an inside view of the building users' and local people's requirements and their associations with the Saddar Bazaar area and the buildings from the British period. Among the questions asked were the following: do you think that these British period buildings are heritage? Based on their reply, they were asked to explain why they thought they were or were not. Do you have any emotional, social or cultural attachment to your building and the Saddar Bazaar Quarter? Would you like to save these structures from demolition? Do you think the heritage status of your building is a good initiative? Has that step have helped you make changes as per your living or business requirements? Is there any government support or are there any guidelines given to you for saving or preserving these buildings? Are you open to the idea that your building still has heritage status but you can change or modify the structure according to your requirements? These questions were the outcome of the literature review on the historical urban and architectural development of Karachi from 1839 to the present day [12,15,18,19,28,30–33] and the 2019 interviews with architects and heritage experts (see Table 1 above), also guided to compile these questions and conduct interviews. Key elements of this visit and the discussions were the personal observations of the area and interaction with the users.

*1.2. Selecting Criteria for the Saddar Bazaar Quarter as the Study Area*

Since its inception in 1839, as the first area to be developed by the British, the Saddar Bazaar has played an essential role in Karachi's social, cultural, economic and urban development. It started as a shopping area with some temporary tent shops, but it became a meeting hub for European officers and their families to enjoy and relax in the bars, cinemas (in the 20th century), and shops, providing all the local and European amenities. In addition, after 1880, when the native population were permitted to move to British-developed quarters, the Saddar Bazaar became a multi-cultural centre and a place for the native population to build their permanent buildings for business and residential purposes. The addition of grain and cloth shops in the Bohri Bazaar area of the SBQ and the construction of the Empress Market in 1889 made the Saddar Bazaar Quarter a point of economic development in British Karachi, which continued even after the independence from the British. This historic urban landscape of the bazaar and its involvement in the city development through its tangible and intangible qualities make it a perfect area to be selected for the in-depth study (Figure 1).

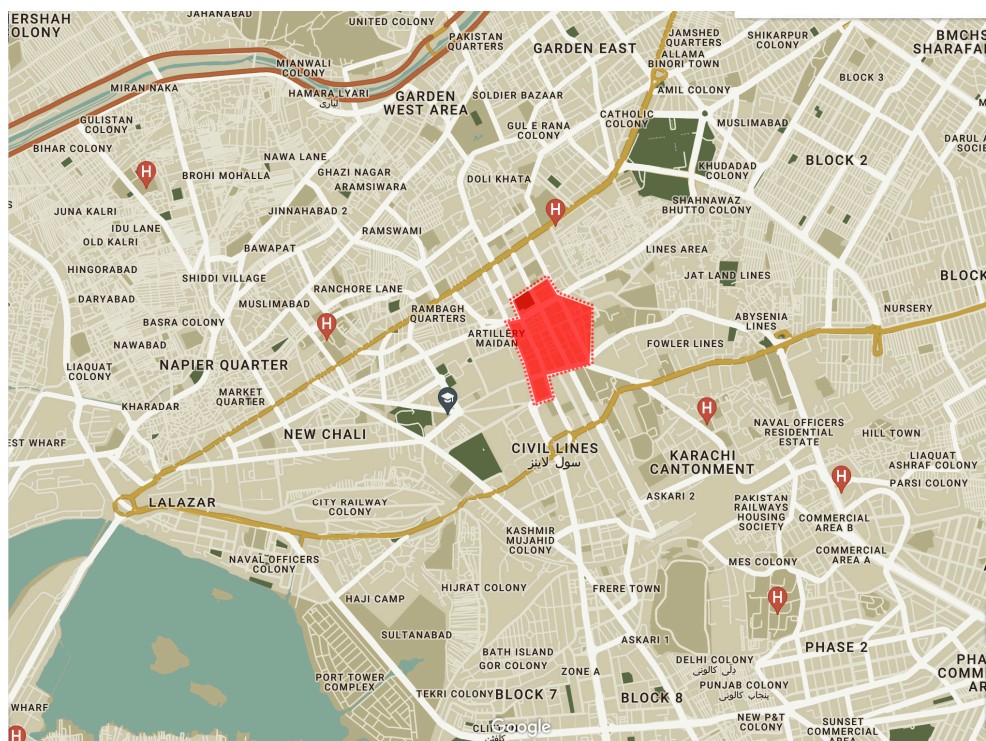

**Figure 1.** The present map of Karachi, with the Saddar Bazaar Quarter marked in red. Source: base map from Google maps, addition of new layer of information by Naveed Iqbal (second author).

## 2. Literature Review

### 2.1. The Historical Origins of Karachi until 1839

Historians [19,34,35] associate the city's origin with Alexander the Great's conquest of South Asia, around 326 BC. Admiral Nearchus, a close friend of Alexander, led a fleet of his army back home from the Krokala harbour, which is thought to be near present-day Karachi. The second location, believed to be the exact present-day Karachi, or near to it, is the city of Debal, from where the Muslim conqueror Muhammad bin Qasim entered Sindh in 711 AD [36]. However, the most important 'start' dates from the early 18th century, when, in 1728, some merchants (mostly Hindu by religion) moved from Kharakbandar (also known as Karack Bunder) to a nearby fishing village. Kharakbandar was a small late seventeenth and early eighteenth-century port, forty kilometres west of present-day Kar-chi. The merchants moved to the fishing village, a small settlement of about twenty-five huts belonging to local fishermen, as the harbour was silted up and could no longer be used. The location of the fishing village is associated with the present-day 'Old Town of Karachi'. Therefore, after moving here, the merchants built a fort in 1729, around the settlement, with a sixteen-foot-high wall made of mud and timber. The fort wall had gun turrets for protection and two gates to enter the fortification area. The one facing the sea to the west was known as Khara Darwaza, and the other towards the Lyari river to the north-east was known as Mitha Darwaza [16,17,19,22,23].

Until the middle of the 18th century, Kolachi or Crochey town (present-day Karachi) had become more of a trading centre, due to its port activity which attracted many rulers to take hold of the city. During the 18th century, the ruling hands changed several times, between the Khan of Kalat and the rulers of Sindh. In 1794–95, after several attacks, the Talpur rulers of Sindh took over Karachi [19,23]. During that time, the increasing port activities and its strategic location also raised the interest of the British in the city. In 1774, the British East India Company sent Lt. John Porter to visit the fortified mud town during his exploratory voyage to the Indus Persian Gulf to explore the city. He described the town as being situated one mile from the creek inside fortified mud walls, with two old, dilapidated Muscat-made guns mounted on round towers. Henry Pottinger also visited

the Kurachee in 1809 as a representative of the East India Company, and found the town in a similar condition as that reported by Porter in 1774 [18,37].

The year 1799 is marked as a landmark, and is when the British showed keen interest in Sindh and Karachi, due to its port. Under the command of Lord Wellesley, in 1799, a political commission was sent to the Talpur rulers of Sindh. After protracted negotiations, the Talpurs granted the British East India Company permission to open their factories at Karachi and Thatta. Thus, the British successfully precluded the other Europeans from starting trade in the region [18,19,37], and in 1839, the British took over 'Kolachi or Kurrachi (later Karachi)' from its Talpur rulers, becoming the city's rulers for the next century [16–18], (Table 3).

**Table 3.** Important historical dates and events in Karachi's history. Compiled by the first author, based on the literature studied.

| S. NO | Milestone Year | Historic Importance |
|---|---|---|
| 01 | 326 BC | Alexander the Great's fleet used the Krokala harbour to go back home. |
| 02 | 711 AD | The exact present-day Karachi, or near to it, is the city of Debal, from where the Muslim conqueror Muhammad bin Qasim entered the Sindh. |
| 03 | 1728 | Some merchants moved to the present-day old town of Karachi. |
| 04 | 1729 | Construction of a fort around the new settlement. |
| 05 | 1774 | The British East India Company sent Lt. John Porter to explore the fort. |
| 06 | 1794–95 | Talpur became the ruler of Karachi. |
| 07 | 1799 | The British East India Company opened factories in Karachi and Thatta, Sindh. |
| 08 | 1809 | Henry Pottinger visited the mud fort town of Karachi. |
| 09 | 1839 | The British occupied the city. |

### 2.2. The British Era of Karachi and Its Evolution (1839–1947)

In 1839 Karachi was a small mud-fortified town with 15,000 people living in the old town inside the fort walls and some of its suburban area. However, the British took over the city of Karachi, due to the port activity of that small town. They saw substantial opportunities, due to its strategic location on the crucial maritime trade route from east to west. The urban layout and architecture of the old town were based upon organic user-laid patterns with narrow, irregular streets with mostly religious buildings used as gathering points for social and cultural activities. Their structures were simple, and based on local wood, mangrove marsh, and stone materials. The houses were built of sun-dried mud bricks with stone foundations, and had flat roofs along windcatchers to allow air and sunlight into the buildings, (Figure 2) [16,25,27].

In 1839, the British troops did not settle in the old town. Instead, they set up temporary tents one mile from the fort, near the Rambagh area (a native sub-urban quarter). Later in the same year, they moved one mile northeast of the Rambagh area, to establish a cantonment area and a bazaar for their troops (Figure 3). Initially, some temporary shops were laid out in the area just beside the cantonment to provide basic amenities for their European officers. The area was given the name 'Suddar Bazar', later called 'Saddar Bazar Quarter' [19,22]. This step became crucial in expanding Karachi beyond the walls of the old town area. For establishing the urban layout of the new area, the British did not follow the indigenous organic form of the old town; instead, they introduced European urban patterns for their area. The early British structures changed to typical colonist look-alike barrack buildings topped by pitched roofs supported by trusses, with the top layer of terra cotta tiles. However, the technique and material were mainly the same as in the indigenous town, such as sun-dried bricks with stone in the foundations [16,26,27]. In the early days of the British in Karachi, the native population did not accept them as the new authority, and there were strong feelings of hostility. To face this resistance and to bring the native population

closer to the British, Seth Naomal Hotchand, a local Hindu merchant and British supporter, started his business in the Saddar Bazaar Quarter, by constructing four shops [19,22].

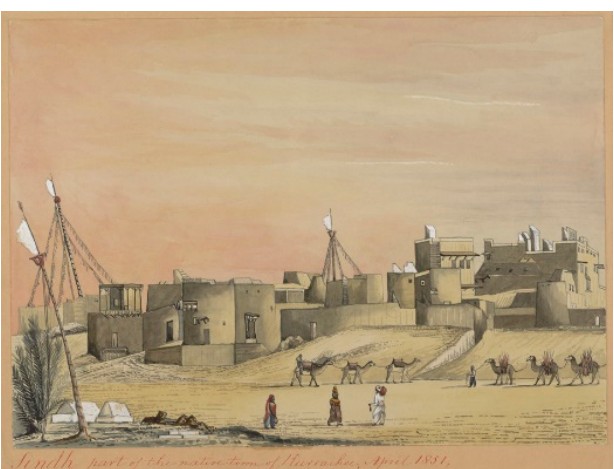

**Figure 2.** Mud Houses of Pre-British Karachi. Painting by Henry Francis Ainslie, 1851. Source: British Library, shelf mark: WD2072; item number: 2072.

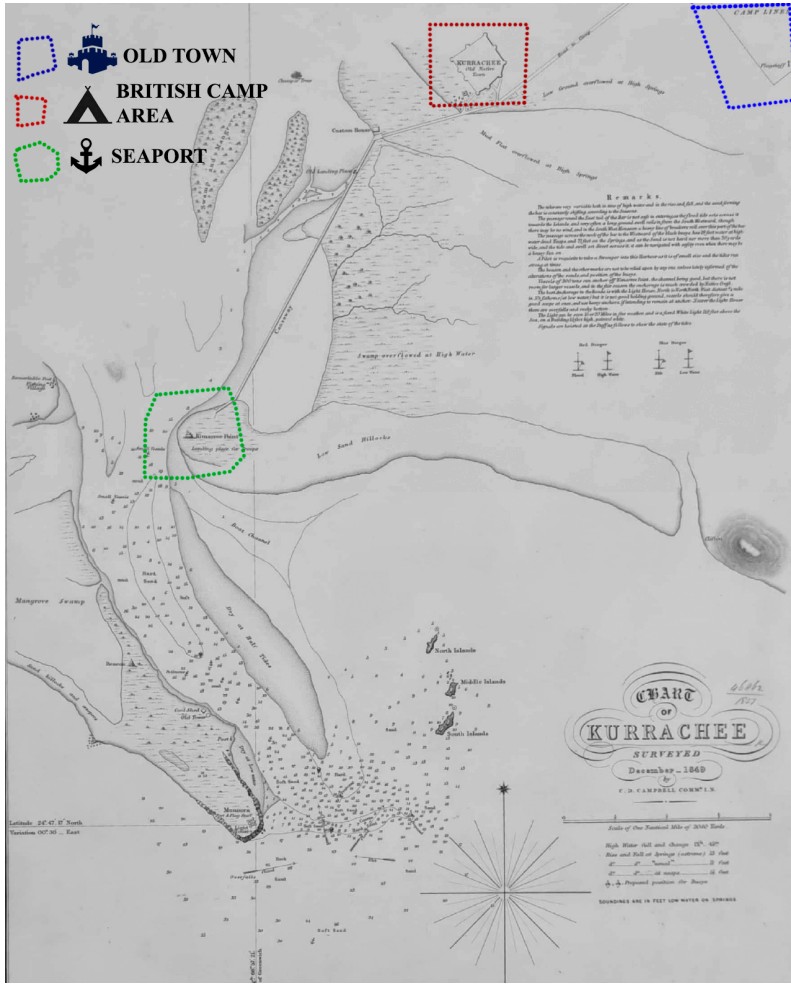

**Figure 3.** 1849 map, showing the location of the Old Town of Karachi, the Seaport, and the Camp area of the British army. Source: British Library, shelf mark: Cartographic Item Maps 147.e.19, UIN: BLL01004869133. Addition of new layer of information by Naveed Iqbal (second author).

The significant development of Karachi started in 1843, after Karachi was inaugurated as the provincial capital of Sindh, and Sir Charles James Napier (1782–1853) was commissioned as the first governor of Sindh. During the Napier period, the British started building office buildings, bungalows, clubs, and barracks for troops and officers. Napier also initiated the development work on Karachi port [26,38]. After Napier was Sir Bartle Frere's tenure (1815–1884) as Commissioner of Sindh (1851–1859), and his plans for Karachi were very ambitious. Frere worked on the urban expansion of Karachi and its road layouts, and commissioned the railway scheme in the city to connect the port to the inland cities. As a result, many British companies opened their offices and warehouses in Karachi, due to the improved port activities. With the increased economic opportunities, many people moved to Karachi, and the city emerged as a growing financial hub, with a population of 57,000 by 1856 [12,23,24,38]. Under British rule, this urban and economic boost gradually changed the construction material from mud bricks to stone [19,24,27]. After the 1857 rebellion, the native army of Indians lost against the British East India Company. It was the beginning of an era that changed the edifices of the city of Karachi and marked the presence of the British as the city's ruling elite. The British government became politically entrenched in Karachi, and they established new areas (quarters) for themselves using European gridiron planning, with functional segregation in every new quarter [14,20,30] (Figure 4).

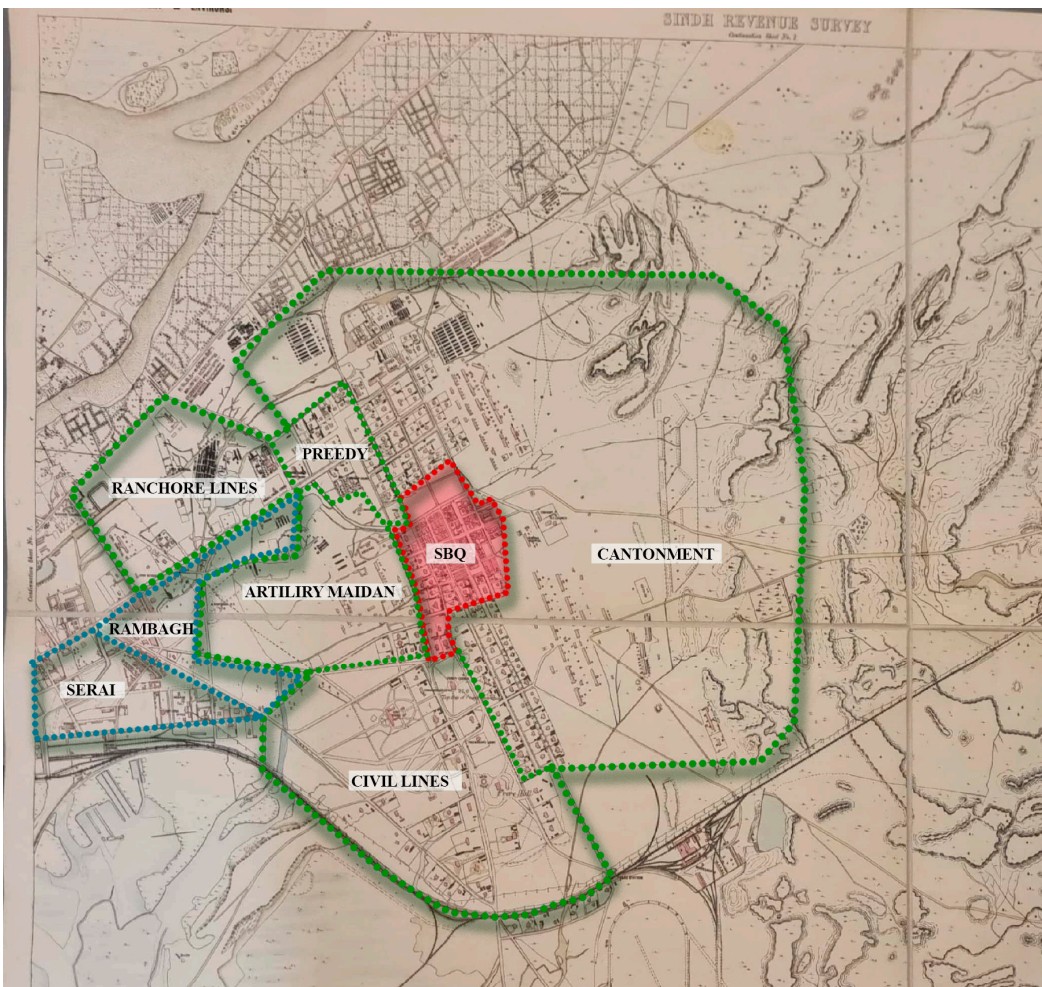

**Figure 4.** 1869–70 map, showing the location of Saddar Bazaar Quarter (Suddar Bazaar at that time) as a central point for the British-developed European and native quarters. Source: British Library, shelf mark: Cartographic Item Maps I.S.100, UIN: BLL01004869127. Addition of new layer of information by Naveed Iqbal (second author).

Among these was the 'Saddar Bazaar Quarter', one of the first areas which the British started to develop, just after the occupation of Karachi in 1839. The bazaar's purpose was to facilitate the European officers and their families living in and around a small radius, with their everyday life and their shopping and leisure activities, through its shops, bars, and cinemas, from 1839 to 1947 [14,15].

In parallel to urban expansion and development, the British also influenced the city's architectural evolution. European styles such Neo-classic, Neo-Gothic, Neo-Renaissance, Neo-Romanesque, Art Deco, Art Nouveau, and the Palladian styles began to sprout soon after the British occupation of Karachi. Two famous buildings of the 1850s are the 'Napier Barracks' and the Trinity Church, 1855, where buff Gizri stone was used, and both buildings still stand in modern Karachi. The first example of a Neo-Gothic secular building, the Frere Hall in 1865, saw the influence of Augustus Welby Pugin (1812–1852) and John Ruskin (1819–1900), famous British architects of the 19th century (Figure 5). These buildings became role models for the British rulers of Karachi and for the local elites and merchants as they started following the European style [19,24,27].

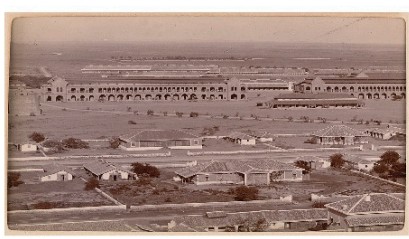 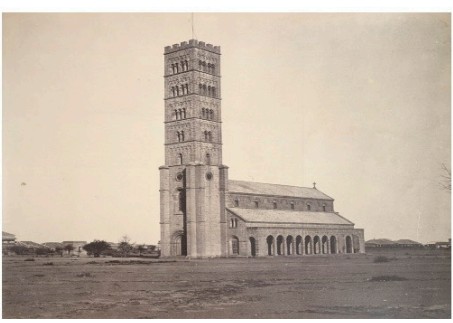 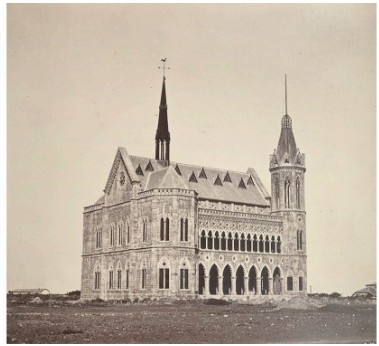

**Figure 5.** L: Napier Barracks photography from 1900. Source: British Library Online Gallery, from album '46 prints titled Karachi Views', shelf mark photo 425/(27), creator unknown. Trinity Church, photograph from 1860, source: British Library Online Gallery, from album "Photographs of India and Overland Route,' shelf mark photo 394/(136), creator unknown. Frere Hall, photographs from 1865. Source: British Library Online Gallery, from album 'Photographs of India and Overland Route', shelf mark photo 394/(137), creator unknown.

In the last quadrant of the 19th century, the British constructed some famous masterpieces in Karachi, including 'The Empress Market, 1889, in the English Gothic pointed style, which Baillie in 1890 also referred to as the 'Domestic Gothic Style' in the Saddar Bazar Quarter', and the Merewether Memorial Tower in 1892, in the pure English Medieval style [18,19] in the Serai Quarter. (Figure 6).

The last two decades of the 19th century saw a rapid increase in the city's population, due to the introduction of a tramway, completing a railway which connected Karachi to the inland cities of British India. The British had also expanded Karachi to 19 quarters, including improved native population quarters, without altering the informal multi-functional quality of the space. The locals also started to accept the British influence on social and cultural life. The government allowed the native population to integrate into the British quarters and convert their temporary business shops into permanent buildings. This gave the native population a chance to start mixing the European styles with the local styles of architecture, which gave birth to the hybridized-classical or 'imperial-vernacular style', or 'colonial architecture'. This style incorporated European classical forms, mostly Italianate, which were considered close to local architectural traditions and blended with indigenous motifs, giving a local charm to buildings. The British rulers also used this hybrid style to construct some government buildings in the 20th century, such as the Karachi Port Trust building in 1916 and the Karachi Municipal Corporation building in 1932, in the colonial hybrid style, depicting the influence of an imperialistic culture on a colonised nation [14,19,27].

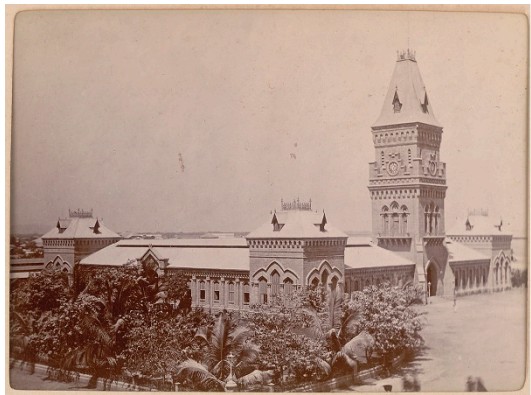
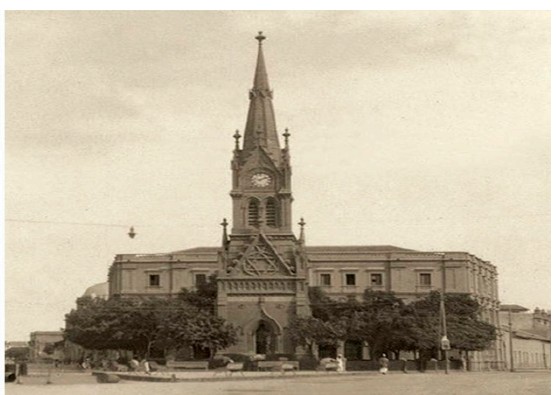

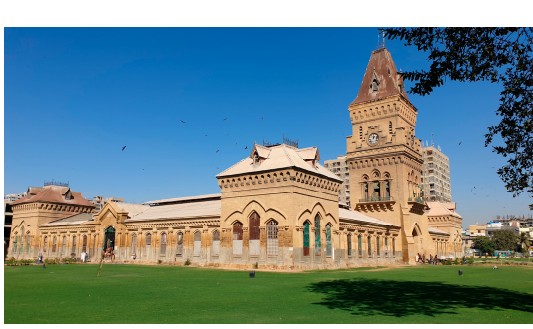
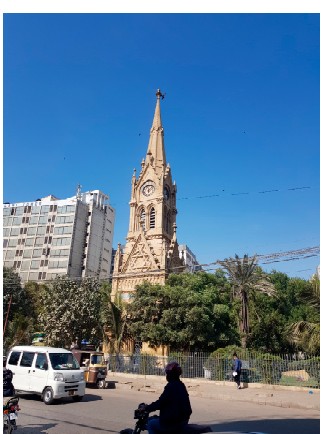

**Figure 6.** Above Left: Empress Market, 1900, from British Library Online Gallery, from album '46 prints titled Karachi Views', shelf mark photo 425/(19), creator unknown. Below Left: Empress Market, picture by Syed Hamid Akbar (first author), 2020/21. Above Right: Merewether Memorial Tower, 1900; the Endowment Fund Trust for Preservation of the Heritage of Sindh (EFT), 2019. Below Right: Merewether Memorial Tower, picture by Naveed Iqbal (second author), 2020/21.

When the native population running the shops, bars, and clubs in Saddar Bazar started moving into the SBQ, they used the colonial architecture style for their buildings. The buildings were usually multi-functional; the ground floor was primarily commercial, and the locals used the upper floors for residential purposes. The Mohammad Ali building, the old Ilaco house, the Jahangir Kothari Mansion, and the Nusserwanjee building in the Saddar Bazar Quarter are some examples of 'colonial style' buildings by the native population from the late 19th century to the earlier 20th century in SBQ (Figure 7).

This resulted in the conversion of the Saddar Bazaar into a multi-class and multi-ethnic cultural hub of the city, where the British and the native population, usually the elite locals, used to come for shopping and relaxation. This norm continued until the 1947 independence, when the British rulers left the city [19,23].

### 2.3. Post-Independence British Architecture Timeline

At the time of independence in 1947, Karachi became the capital of a 'new country' (in 1967, Pakistan's capital was shifted from Karachi to Islamabad). The city was confronted with significant socio-political pressure in the form of demographic changes, and the urgent need for buildings to accommodate six million refugees [23,38] and office buildings for the administrative staff of the provincial and federal capital. By 1951, Karachi's population had doubled, and increased to 1.137 million, and since 1972 Karachi's population growth rate has been around five per cent [23,39]. As a result, the city of Karachi went through some significant functional changes that also altered the urban pattern of Karachi. The provincial office buildings of the British period were handed over to the central government, such

as the Sindh secretariat, and the assembly buildings to the central offices, while the Sindh secretariat office was shifted to Napier Barracks and provincial assembly gatherings were organised at Khaliqdina Hall. The Indian migrants settled in the city's open public spaces and buildings which the departing Hindus left behind. Most of these settlements were around the Saddar Bazar, within walking distance. Besides this, the embassies of many countries, such as the United States of America, Japan, France, Brazil, Argentine, Iran, and the USSR (now Russia), were also established in the Civil Lines quarter, within a two-kilometre radius of the SBQ. A new university was also established in 1952, near the Saddar area; the older educational institutions and administrative buildings were just a few kilometres from the Saddar Bazar. All these demographic and functional changes reshaped the city of Karachi, but the Saddar Bazaar Quarter evolved to be a hub for relaxation, leisure, and shopping for various groups that included the diplomats, politicians, artists, intellectuals, government officials and students living and working in or around the SBQ. The site symbolised the British legacy of a social and cultural gathering hub for European and native elite, catering to a multi-cultural, multi-class community of intellectuals, artists, and poets after independence [19,23], (Figure 8).

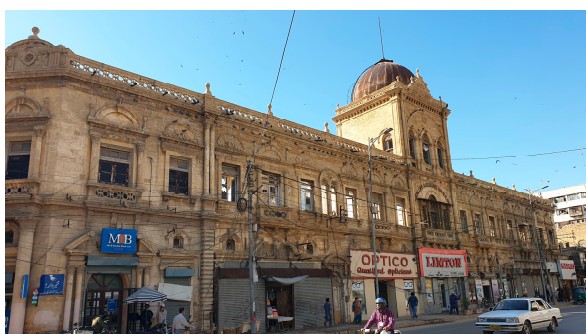
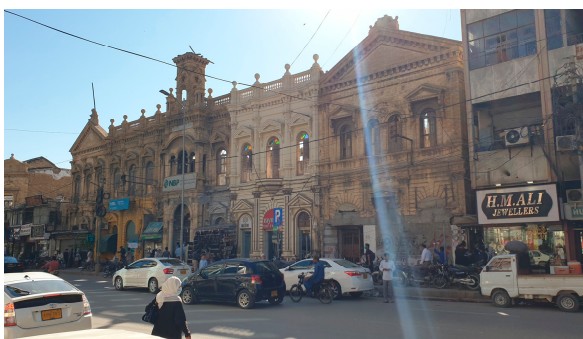
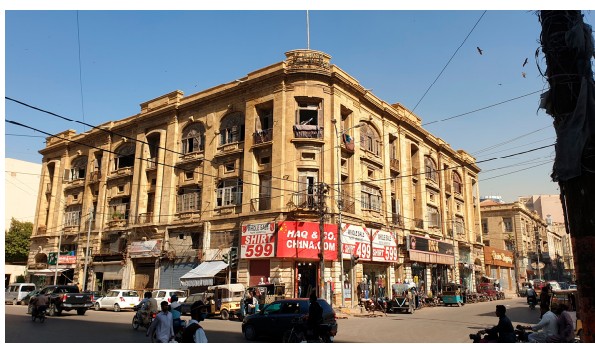
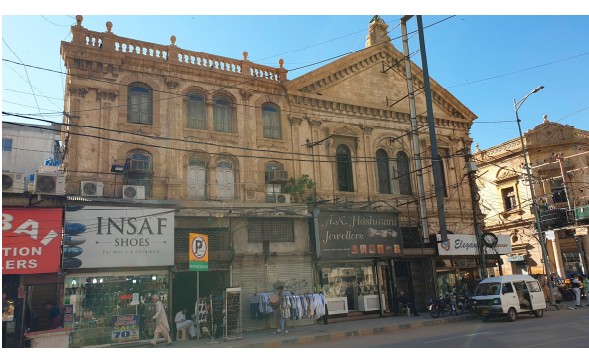

**Figure 7.** (**Above Left**): Old Ilaco House. (**Below Left**): Jahangir Kothari Mansion. (**Above Right**): Mohammad Ali Building. (**Below Right**): Nusserwanjee Building. Source: Syed Hamid Akbar (first author) 2020/21.

In the mid 1950s, the government took some initiatives to move the immigrants west of the old town, along the Lyari River. By the 1960s, the boundary of Karachi had expanded beyond the British city, due to the development of new corporate housing societies around the city, away from the old quarters where the wealthier residents started to settle. During the 1960s and 1970s, trade activities increased at Karachi port, as it became one of South Asia's most prominent and busiest ports. This economic boost attracted many people from the inland cities of Pakistan to Karachi, resulting in an increase in the city population of 217%, compared to 1951 [23,39]. Although in these two decades Karachi was one of the biggest metropolitan areas of south Asia, the increase in population also affected the city. For that reason, more residential, commercial and corporate spaces were required. To accommodate these increasing requirements, changes were made in the building by-laws, which allowed increased building heights and more covered areas, which favoured the land

developers and real-estate mafia in Karachi. They preferred to construct new buildings with more floors and an increased covered area, instead of sticking with the old historic buildings that offered less covered area and limited floors [23,40]. In addition, these British period buildings were not considered historical, compared to the Mughal-era buildings [15], and the laws which Pakistan adopted to deal with heritage and culture were based upon the 'Ancient Monument Preservation Act, 1904' from the British period, which changed in 1968 and then again in 1975, for better implementation and protection of old buildings [31,41]. However, due to unclear definitions, it was mainly applied to archaeological sites or single monumental buildings, and did not cover the protection of the British-period buildings, which encouraged land developers and building owners to demolish and replace the old buildings with new mid-rise plazas in the historic quarters of Karachi [20,31,42,43]. The Saddar Bazaar Quarter was one of the worst-affected areas, as its literati, artists, poets, and other elite users also moved to the newly developed area. The Saddar Bazar became just a 'bus transit camp' for people moving to their workplaces in the historic quarters from their new houses. This led to its environmental degradation, along with urban and physical deprivation. The area started losing its bars, cinemas, bookshops, and places of social and cultural gatherings to business activities and storage, and this marked the end of the Saddar Bazar as a city centre [12,14,15,23].

### 2.4. Legal Efforts and Protection of Historic Buildings in Karachi

In the 1970s, the government tried to preserve some of the monumental heritage buildings of the British era. However, the lack of technical knowledge of the materials, construction, and craftsmanship resulted in losing some historical layers from the buildings' fabric, as in the case of Trinity Church and Frere Hall. The original roof of Trinity Church was replaced with a barrel vault roof, and in the case of Frere Hall, the dormer windows from the ceiling were permanently removed (Figure 9).

In response to these unacceptable alterations, some architects and activists, and some private organisations started to raise their voices to protect British-era buildings in Karachi [20]. The outcome of their struggle was that a private organization (an NGO, founded by Ar.Yasmeen Lari and Suhail Zaheer Lari, with a focus on heritage and humanitarian aid) was able to preserve the historic Flagstaff House(now known as Quaid-e-Azam House Museum). The building was designed by the famous British architect Moses Smoke (1875–1947), and is considered as one of the first building designed by him. He designed the building for a Parsi merchant, and it was completed in 1890. The British Indian army rented the building for their senior military officers. In 1944 Muhammad Ali Jinnah (the founder of Pakistan) bought the building for his personal use as a house. Based on its historical values, in 1985, the Heritage Foundation, Pakistan, preserved and converted it into a museum with all the belongings of Muhammad Ali Jinnah's life [19,20,44]. This also resulted in initiating a list of 44 important buildings that came under the supervision of the 'Karachi Development Authority' and the 'Department of Archaeology and Museums' under the 'Karachi Building and Town Planning Regulations of 1979', to give them protection. [20,45]. These buildings were considered important historic buildings, based upon their monumental scale and architectural/historic values [20]. However, it was in 1994 when a separate act with the title 'Sindh Cultural Heritage Preservation Act of 1994' was passed [46]. Under the act, initial inventories were made from 1995–1997, listing 581 historic buildings from nineteen historic quarters of Karachi that were given protected heritage status [19–21]. The Saddar Bazaar Quarter was also among the nineteen quarters, and 76 buildings of British period were included in the listing for protection, based on their historical values [19–21,29]. The buildings were of different typologies, such as residential, commercial, residential-cum-commercial, or commercial-cum-residential buildings (Table 4), including two monumental buildings, the Edulji Dinshaw Dispensary and the Empress Market. This was a new concept for Pakistan, specifically in Karachi, where until now, only a few monumental masterpieces were considered buildings of historical value [20,29].

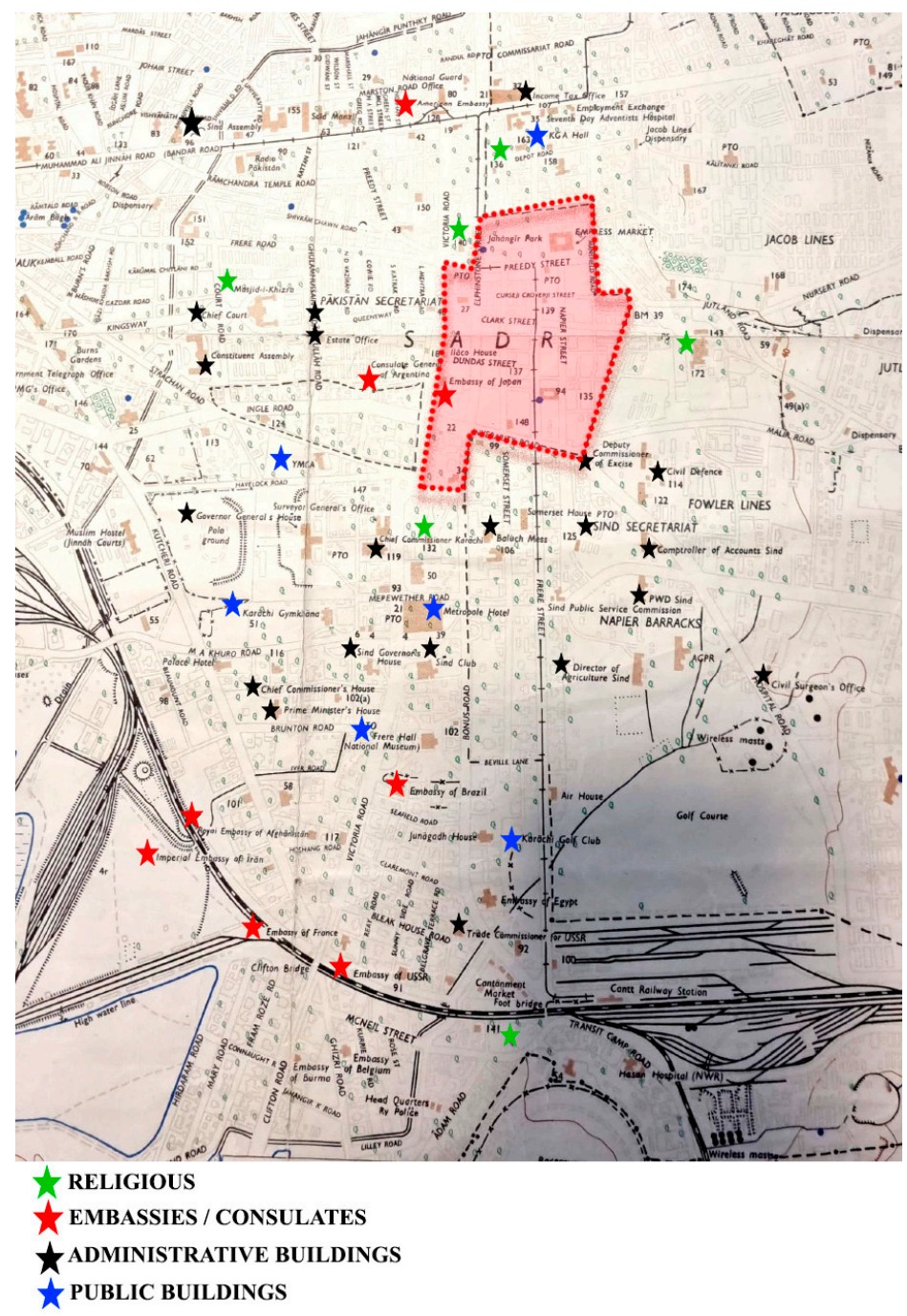

**RELIGIOUS**

**EMBASSIES / CONSULATES**

**ADMINISTRATIVE BUILDINGS**

**PUBLIC BUILDINGS**

**Figure 8.** 1954 Map of Karachi shows different important administrative, diplomatic, public and religious buildings around the Saddar Bazaar Quarter. Source: British Library, Main Catalogue: system number, 019173232. Addition of new layer of information by Naveed (second author).

### 2.5. Post-Enlistment British-Era Heritage in Saddar Bazaar Quarter, Karachi

The 1994 Act and Enlistment were supposed to protect the heritage of Karachi from demolition, but proved less effective and remained insignificant. The 1994 Act did not mention any laws or guidelines on preserving or altering the buildings according to changing user and urban requirements. The act also did not offered any incentives to the building owners against the protected status. In addition, the process of obtaining approval to make changes in protected buildings was lengthy, with the limitation of the covered floor area. In contrast, the by-laws for constructing new buildings allowed the heritage-property owners to have more built-up areas and more floors. These conflicts between different laws,

and the mismanagement and miscommunication between various departments supported the building owners to apply to other departments to delist their properties [21,32,43].

**Table 4.** Showing typological division of 76 buildings of the SBQ that were given protected heritage status in 1995–1997, compiled by Syed Hamid Akbar (first author). Based on Karachi Heritage Buildings Re-survey Project of 2006–09, Heritage Cell, Department of Architecture and Planning, NED University, Karachi (Heritage Cell, DAPNED) [29].

| Typology | | | | | | |
|---|---|---|---|---|---|---|
| Residential | Commercial | Residential, Commercial | Commercial, Residential | Amenity and Religious | Institutional | Total |
| 03 | 22 | 03 | 42 | 05 | 01 | 76 |

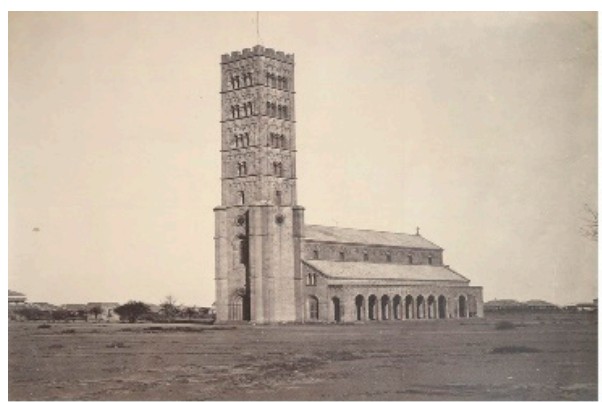
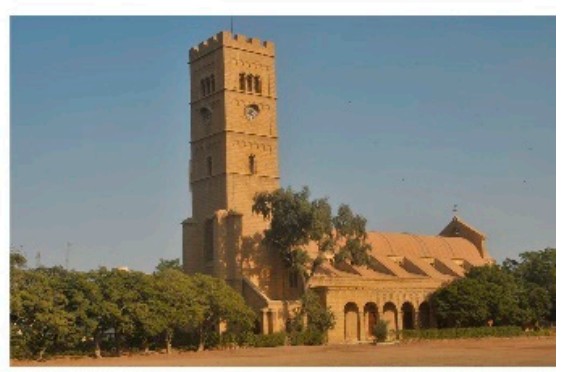
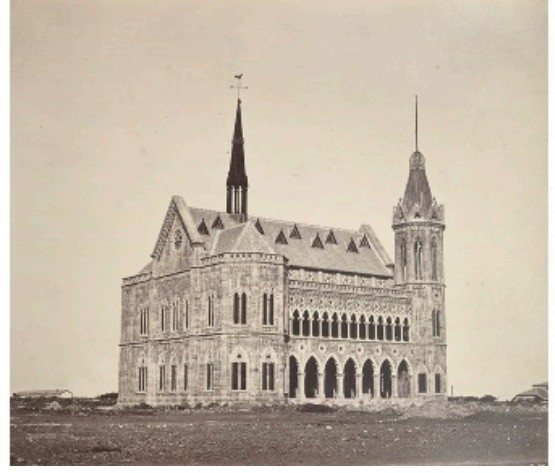
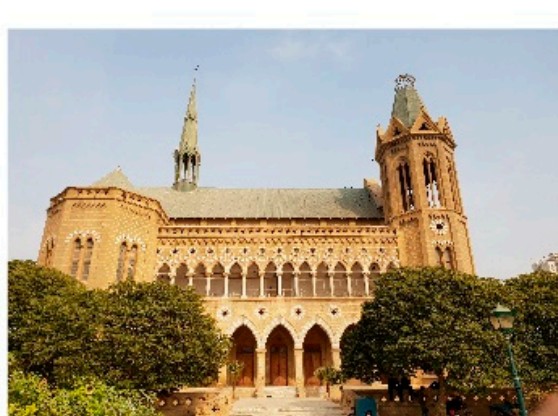

**Figure 9.** (**Above Left**): Trinity Church, photograph from 1860, source: British Library Online Gallery, from album 'Photographs of India and Overland Route', shelf mark photo 394/(136), creator unknown. (**Below Left**): Frere Hall, photographs from 1865. Source: British Library Online Gallery, from album 'Photographs of India and Overland Route', shelf mark photo 394/(137), creator unknown. (**Above Right**): 2001 image of Trinity Church. Source: Lari and Lari, 2001. (**Below Right**): present image of Frere Hall, Source: Syed Hamid Akbar (first author), 2020/21.

In some cases, they neglected the maintenance of protected buildings, leading to deterioration or even demolition [15,32,43]. This resulted in the decline in the city's historic urban landscape and, eventually, the destruction of heritage buildings, giving preference to other commercial and economic gains; just after the 1995–97 listings, this was the primary reason behind the demolition of some buildings in Karachi [20,43]. The SBQ and its heritage buildings also followed the same fate, and buildings such as the 'Olympia Building

(1995-070[1]) and 'Medina Building (1995-071[i]), with no record of delisting [33]. While 'Palia House (1995-065[i]) was given protected status in 1995, the 'Karachi Building Control Authority' declared it dangerous in the same year, and it was demolished [16,17,19,24]. A survey conducted by the Heritage Cell at NED University, Karachi, from 2006 to 2009[2], documented and reported that of the 76 buildings given protected heritage status through the listing of 1995–97 in the SBQ, only fourteen per cent were in a good state[3] of condition. At the same time, thirty-six per cent faced a high degree of threat[4] [29]. (Figure 10).

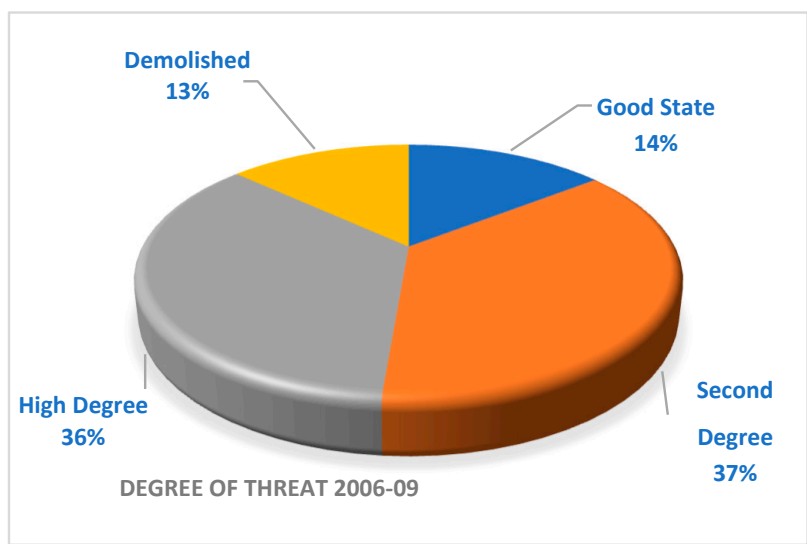

**Figure 10.** Compiled by Syed Hamid Akbar (first author), showing the percentage of buildings according to the degree of threat, based on Karachi Heritage Buildings Re-survey Project 0f 2006–09, Heritage Cell-DAPNED [29].

## 3. Research Fieldwork and Findings

The literature review helped to understand Karachi's historic urban and architectural evolution and development. The literature analysis shows that the British rulers of Karachi converted a small fishing town into a world mega city and South Asia's biggest port. and that when the British departed they left a handful of tangible and intangible memories of the city's journey in the form of its urban layout, architecture, landscape, and culture.

As the first step in fieldwork, different archival departments were visited to look for historical maps, pictures, written documents, and newspaper articles, etc. The historical maps and photographs from various archival departments in Karachi, Pakistan, and the data from the British Library, London, were useful in understanding the previous literature. Archival data supported the data found through the literature analysis, and became helpful for fieldwork and documenting the 76 buildings in the SBQ.

### 3.1. Findings—Physical Condition of Buildings

From 1995 to 1997, 76 buildings were given e protected status under the 'Sindh Cultural Heritage Preservation Act of 1994'. To find the present condition of these buildings and investigate the reasons behind their current state, research visits were carried out in January 2019, and again from December 2020 to January 2021, to collect data and document the SBQ. The analysis of fieldwork data shows that the SBQ historic buildings which had protected status or had protected status are on the decline. Compared to the 2006–09 report by the Heritage Cell-DAPNED, the number of buildings in good condition has decreased, and the buildings with a high degree of threat and the ones which have been demolished have increased (Figure 11).

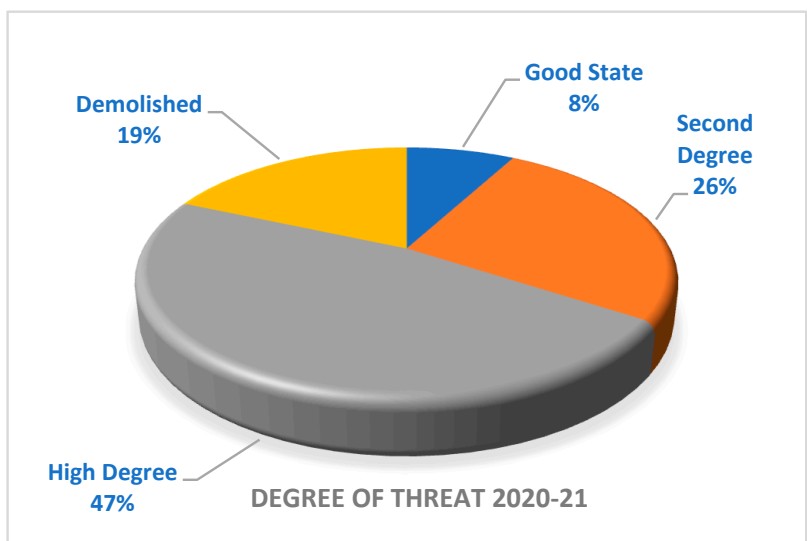

**Figure 11.** Based on the research fieldwork by Syed Hamid Akbar (first author), showing the percentage of buildings according to the degree of threat in 2020–2021.

The comparison of the 2006–09 report by the Heritage Cell-DAPNED and analysis of fieldwork data by the first author also shows that the commercial and commercial–residential typologies are the most affected among the different typologies of buildings. These buildings' conditions have worsened in the last decade. Many buildings from these two typologies that were in good condition or just facing second-degree threat are now on a high-degree-threat list or have been demolished. (Table 5).

Buildings such as the Jahangir Kothari Mansion (1995-002[i]), 'Krishna Mansion (1995-003[i])' and Fazal Manzil (1995-016[i]) from commercial–residential typology have moved to a high-degree-threat state. In the latter two buildings, only the ground floor is used for commercial purposes, while the upper floors have lost their residential function and are now vacant and facing partial demolition (Figure 12).

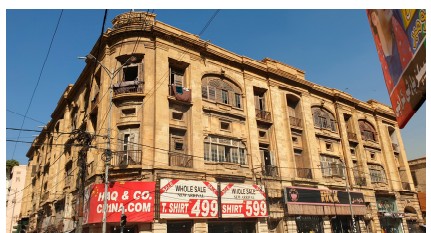 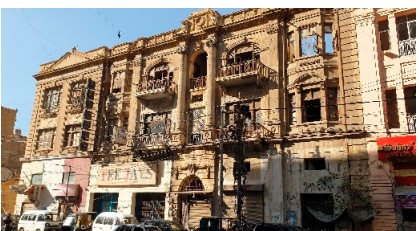 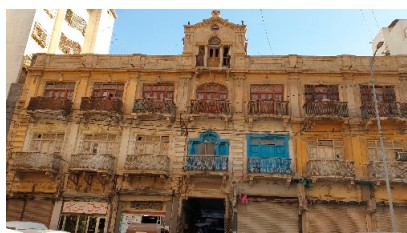

**Figure 12.** L–R: Jahangir Kothari Mansion, Krishna Mansion, Fazal Manzil. Photographs by Syed Hamid Akbar (first author), in 2020–2021.

Some other buildings, such as 'Nabi Manzil (1995-020[i])' and 'Braganza House (1995-073[i])', which were in a good state of condition in the 2006–09 report, are now empty, without any function and facing a high degree of threat in 2021. For both buildings, it was found that the split ownership after the passing of the main owner and the issue with the inheritance laws are the main reasons for their current bad condition. Both buildings are entirely vacant, and some floors have even fallen or started to deteriorate (Figure 13).

**Table 5.** Comparison based on the Degree of Threat of different typologies from the SBQ from 2006–09 survey by Heritage Cell-DAPNED and the research fieldwork(data in bold text) by Syed Hamid Akbar (first author) in 2020–2021.

| Degree of Threat | | Typology | | | | | | Total |
|---|---|---|---|---|---|---|---|---|
| | | Residential | Commercial | Residential, Commercial | Commercial, Residential | Amenity | Institutional and Religious | |
| Good State | 2006–09 | 01 | 01 | - | 05 | 04 | - | 11 |
| | **2020/21** | **00** | **01** | **-** | **03** | **02** | **-** | **06** |
| Second-Degree Threat[5] | 2006–09 | 01 | 06 | 01 | 18 | 01 | 01 | 28 |
| | **2020/21** | **00** | **07** | **01** | **09** | **03** | **01** | **21** |
| High-Degree Threat | 2006–09 | 00 | 13 | 00 | 14 | - | - | 27 |
| | **2020/21** | **01** | **11** | **01** | **22** | **-** | **-** | **35** |
| Demolished | 2006–09 | 02 | 02 | 01 | 05 | - | - | 10 |
| | **2020/21** | **02** | **03** | **01** | **08** | **-** | **-** | **14** |
| Total | 2006–2009 | | | | | | | 76 |
| | **2020/21** | | | | | | | **76** |

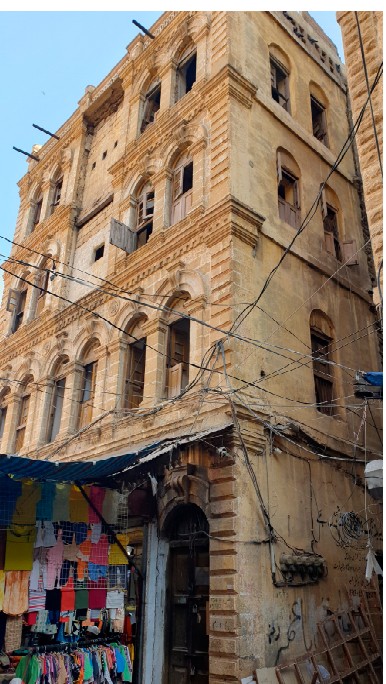 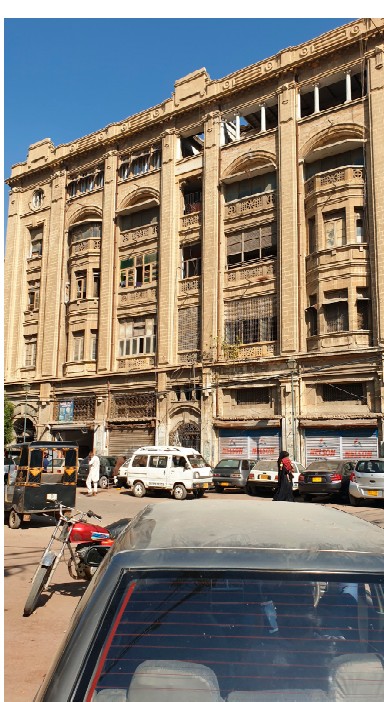

**Figure 13.** L–R: Nabi Manzil, Braganza House. Photographs by Syed Hamid Akbar (first author), in 2020–2021.

A lack of interest on the part of the respective authorities dealing with the heritage of the Karachi and Sindh province (the Sindh Building Control Authority, and the Sindh Culture Department) in taking any measures and solving the discrepancies in different laws to protect these buildings, is also causing ignorance among owners. The buildings are poorly maintained and face a high degree of threat. Some buildings were even found to be in a much worse condition during the fieldwork of 2020–21, in comparison to the 2006–09 report, in which they were placed at a high degree of threat. These buildings have lost the upper floors, and the facade material is deteriorating and falling, making them a constant danger for the Saddar Bazaar Quarter users. The 'Sheikh Fida Ali Building (1995-048[i]) is one of the buildings in this condition. The façade of the building facing the Mochi Gali (Street) is tilting outward, with the wood cladding missing from some sections and exposing the bricks to the external conditions. Inside the building, the upper floors have partially collapsed, meaning that the upper stories of the building are empty. Only the ground floor is used for commercial purposes, with hardware, clothing, and daily-use-items shops (Figure 14).

The owner of the building, who runs the hardware shops on the ground floor, pointed out the ignorance of the authorities concerned and the fact that they were not providing any proper guidelines or help with preserving/restoring the partially demolished upper floors and using them for new functions. The owner of the Sheikh Fida Ali building also mentioned that the building is in such bad condition it could collapse at an time or could be demolished. From Figure 14, we can see that modern brick and concrete have replaced the neighbouring buildings, and the present condition of the Sheikh Fida Ali building is also moving towards the same fate. Rainbow House (1995-023[i]), situated at the junction of Zaibunnisa street and Albert street, is even worse than the Sheikh Fida Ali buildings. The adjacent shop owners and some hawkers around the shop indicated the abysmal condition of the building façade. They said that the wood pieces from the façade had fallen into the street, making the buildings very dangerous for the people around the building, but luckily no one was injured. However, some shops on the ground floor are still in use (Figure 15).

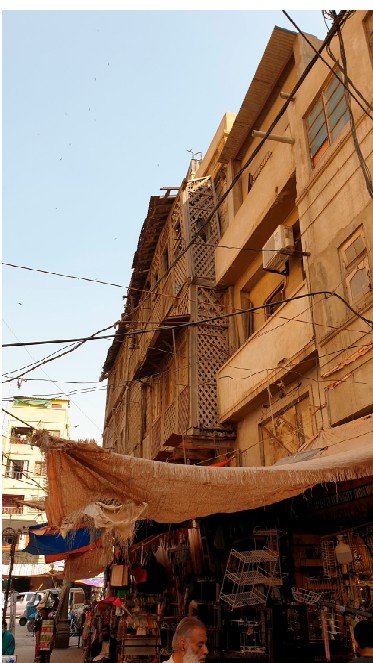 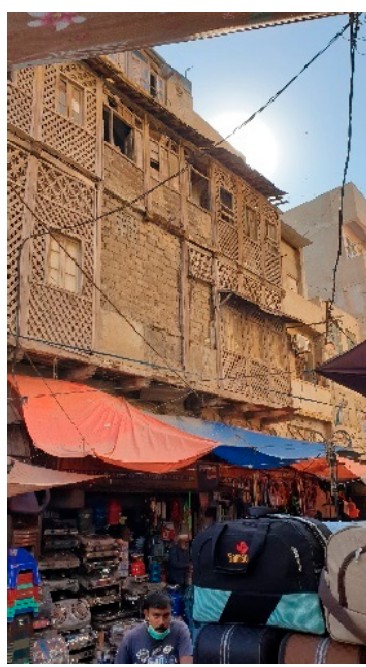

**Figure 14.** Sheikh Fida Ali Building. Photographs by Syed Hamid Akbar (first author), in 2020–2021.

Some buildings are in worse condition, as described in the definition of high-degree threat. In the Victoria Furnishing building (1995-013[i]), only some of the building's façade stands. In the case of Captain House (1995-057[i]), even the facade is gone, and just some ruins of the building remain (Figure 16). In both cases, it can be seen from Figure 15 that the neighbouring sites have newly built structures. This implies that these two heritage buildings could soon be replaced by newly built structures, to fulfil the users' social, cultural and economic requirements. Besides these privately owned buildings, many government-owned buildings were also facing a high degree of threat, such as the Empress Market (1995-047[i]), which was also in a good state in the 2006–09 report. The building is in a partial state of deterioration, with many holes in the roof of the building. The fieldwork shows that many buildings from the 1995–97 listing are standing and holding their place in the historic urban landscape of the SBQ, but they are slowly losing their existence, for various reasons.

*3.2. Findings—Reasons behind the Present Conditions*

During the fieldwork, different stakeholders of the Saddar Bazaar were interviewed, to find out the reasons behind the buildings' current state and officials' ignorance. The people interviewed were architects, heritage activists, conservationists, building owners, and users of the SBQ. Based upon the interviewers' responses and personal observations, the reasons behind these conditions varied from the gaps in the heritage laws, flexible building by-laws, and conflicts among these laws. The process of approving alterations in the protected heritage buildings by the authorities is lengthy, as compared to just demolishing the building and constructing a new building. In addition, during the interactions with the building owners, users and shop owners, it was found that some people have developed personal associations with the building and the area. Some people have family associations, and mentioned that their forefathers passed the building on to them, and they were born and raised in the area. On the other hand, some mentioned the economic association, and that they have been running businesses in the location for a long time. These personal and communal attachments have the potential to become a baseline for the protection of the buildings [47]. In addition, in the SBQ, these collective values have also made possible the continuation of these structures as unchanged or informally reused, according to their requirements.

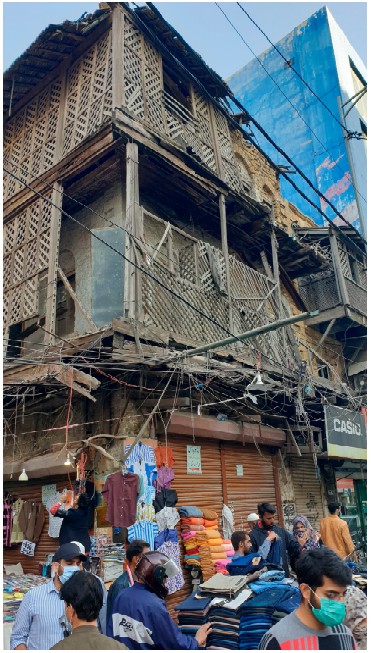
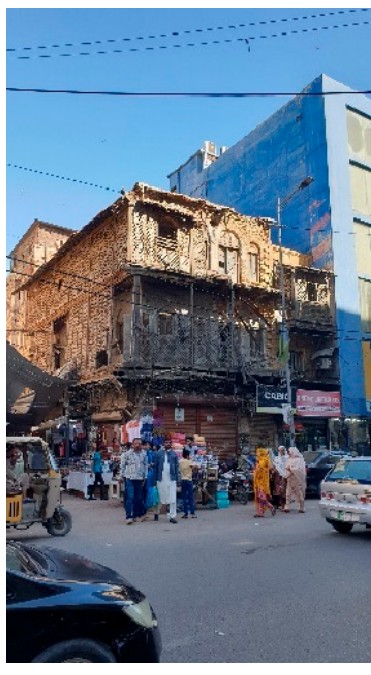

**Figure 15.** Rainbow House. Photographs by Syed Hamid Akbar (first author), in 2020–2021.

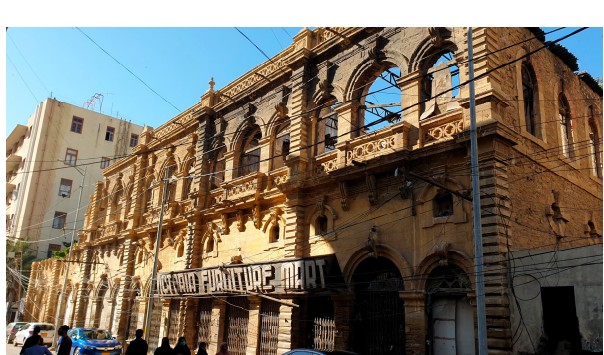
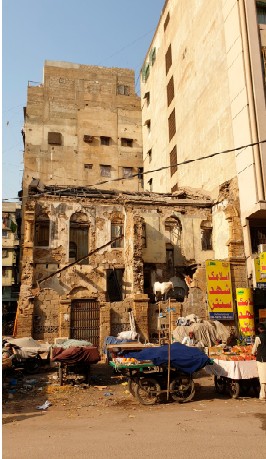

**Figure 16.** Victoria Furnishing Mart, Captains House. Photographs by Syed Hamid Akbar (first author), in 2020–2021.

In addition, the random selection of buildings for protection without any clear criteria for the 1995–1997 listing was a significant reason for some building owner to become scared of the protected status and try to avoid it or remove their buildings from the list [20,28,31].

A group of land developers and building owners benefitted from these conflicts and the gaps in the heritage law and listing criteria/process. They won the case against the government of Sindh in 2017 [48]. This resulted in delisting all buildings with protected-heritage status from 1995 to 2017. The government of Sindh was advised to make proper laws under the 1994 Heritage Act, and to produce new listings of protected heritage [32,48]. Although the re-notification process of listing started again in 2018 after the court order, on the new lists, many monumental buildings such as the Empress Market (1995-047[i]), the Edulji Dinshaw (1995-043[i]) Dispensary, and Jama Masjid Qasaban (1995-058[i]), along with some modest buildings such as Rainbow House (1995-023[i]), the Kanji Kara building (1995-030[i]), and Dossalani Terrace (1995-057[i]) were found to be missing. The authorities have not given an explanation for or a reason behind their exclusion. All these excluded buildings are in bad condition and face a high level of threat. If no proper guidelines are

provided to the building owners or users, soon they will have no choice but to demolish them and build new residential or commercial plazas to comply with their current socio-economic requirements.

Along with the above issue, there are conflicts among shop owners and building owners, regarding rental agreements. Most shops in Karachi's historic quarters, including the Saddar Bazaar Quarter, are rented in the traditional old Pakri System: renting out a shop or building to an individual for an extended period for a fixed amount per month. In the Literature Review Section, many authors and researchers [15,21,32] pointed toward the negative impact of this system on the area's economic development and the present condition of heritage buildings in Karachi. Due to the conflicts in the Pakri System, owners are not able to increase the amount of rent for the shop, and are not in favour of maintaining the physical condition of their building/s, while the people renting the shop and running their business are also not taking responsibility for maintaining the building. They just maintain their own shop. As a result, many buildings are in decline, due to this conflict. During the fieldwork, it was found that many protected heritage buildings in the Saddar Bazaar Quarter face this issue (Table 6).

**Table 6.** Buildings Rented through the Pakri System in Saddar Bazar Quarter—Based on Fieldwork 2020/2021 by 1st author.

| S. No | Building Name | Listing Number | Threat Level 2020/2021 |
|-------|---------------|----------------|------------------------|
| 01 | Jahangir Kothari Mansion | 1995-002 | High-Degree Threat |
| 02 | Krishna Mansion | 1995-003 | High-Degree Threat |
| 03 | Nusserwanjee Buildinng | 1995-010 | Second-Degree Threat |
| 04 | Fazal Manzil | 1995-016 | High-Degree Threat |
| 05 | Abu Building | 1995-026 | High-Degree Threat |
| 06 | Sunderji Hamji Building | 1995-027 | High-Degree Threat |
| 07 | Khyber Hotel | 1995-042 | Second-Degree Threat |

These are just some cases where shop owners and building owners were comfortable sharing the rental details. Among these seven buildings, two are facing a second-degree threat, and the reason for this is that the upper floors of these two buildings are still in use, compared to others where the upper floors are vacant, or even where floors have fallen or are being demolished. The shop owner of the Abu Building (199-026[i]) mentioned that the building owner is responsible for the current state of the building, and that he is just an occupant of the building and running his business. Therefore, he is only accountable for maintaining his shop to attract customers. When the building owner was asked the same question, he replied that he is receiving par low par value of rent from the shops, so he does not want to maintain the building and is more interested in replacing it with a multipurpose, medium-rise building.

This situation is not limited to just the Saddar Bazar Quarter; other historic quarters face the same problem. One example is the Ather Mansion (1997–023[6]) in the Rambagh quarter, which was demolished in 2019 [21]. In addition, the British-era buildings associated with Karachi's industrial development, such as the railway or the Karachi port warehouses, workshops, and bungalows, are still not considered as heritage by the locals and the heritage laws. No particular category/description is specified in the laws drawn up for dealing with and giving protection to industrial buildings in Pakistan. [31,48].

## 4. Results and Discussion and Research Limitations

*4.1. Results*

The study shows that Karachi is in a rapid transition, putting pressure on the heritage of Karachi at a startling rate. The buildings which once witnessed and played a vital role in the socio-economic development of a small mud-fortified town into one of the biggest cities under British Raj in the Indian subcontent are facing the dilemma of urban and economic development and changing user requirements. The Saddar Bazar Quarter and the historic urban landscape, which was once a hub of social and cultural gatherings during the British period and the decade after independence, is also affected and in decline, moving towards a changed and modern urban landscape. Many British-period buildings have been demolished, despite their 'protected-heritage status'. The fieldwork gives a clear image of the present condition of British-period buildings of the SBQ. Many buildings are in bad condition. The area has lost its ensemble quality as a socio-cultural hub of the British period. It has become a place with a disintegrated environment, with economic activities and serving as an inter-city transportation hub, which is making the situation in the SBQ continue to decline.

The map below in Figure 17 is drawn based on the research fieldwork conducted by the first author. It shows the present condition and locations of the 76 buildings from 1995–97 heritage listings. We can see that most buildings are under a high degree of threat but still standing and fulfilling user requirements, typically as shops and houses, and with streets informally accommodating the changing urban and user needs. This means that these buildings can withstand the changing contemporary urban and user requirements and have the potential for survival for future generations. In the past, these buildings played their part in the city's socio-economic development. They can continue to play the same role in the present time with new functions or even the same old function with improved conditions. The current users and owners favour saving these buildings from demolition. Their response is to re-utilise the buildings, but they need guidance and incentives to fulfil their current socio-economic requirements.

*4.2. Discussion*

Based on the fieldwork findings, analysis and results, it is clear that the historic urban landscape of the Saddar Bazaar Quarter is declining, and that its historic buildings depicting colonial architecture and its British legacy are losing their existence for the different reasons we found in our study. However, it is also a fact that the historic urban landscape and the remaining British-period buildings play an important role in generating economic gains. The British-period buildings of the Bohri Bazaar area are the wholesale shops (Figure 18), while the British-period buildings along Zaib-un-Nisa street are still hosting commercial activities, mainly on the ground floor (Figure 19), and the famous Empress Market, although in deteriorating condition and partially maintained is still standing and hosts a lot of daily used-item shops/stalls (Figure 20). This shows that the British-period structures are still playing an active role in the economic generation for the users, the SBQ and the city of Karachi. This means that these structures, individually and on an urban level, can be adopted, reshaped, and reused formally or informally.

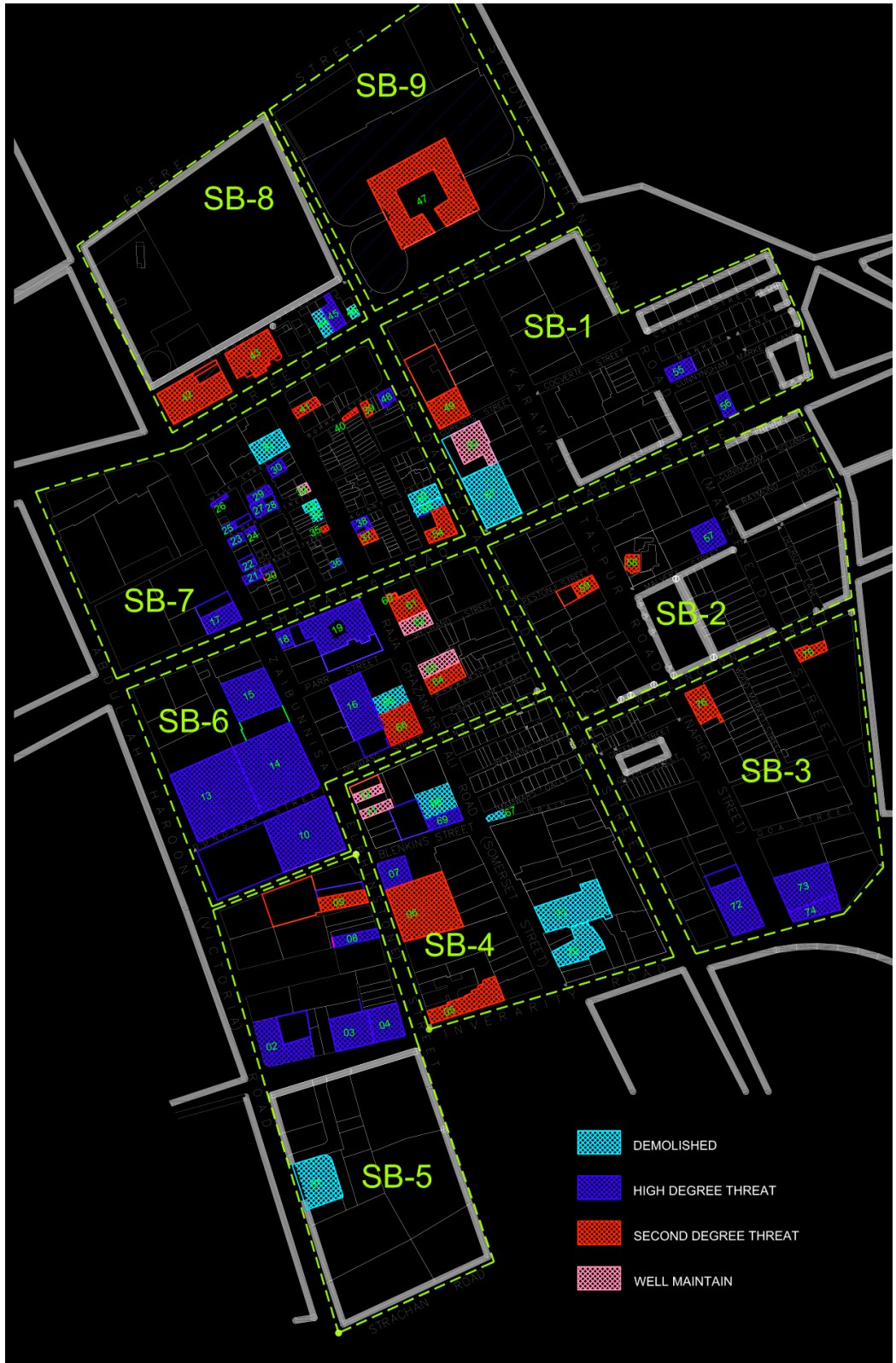

**Figure 17.** Map of Saddar Bazaar Quarter. Based on the research fieldwork by Syed Hamid Akbar (first author) in 2020–21. Showing the present condition and location of 76 buildings from the 1995–97 heritage listing. Source: AutoCAD drawing by Heritage Cell-DAPNED, edited by Syed Hamid Akbar (first author) and Naveed Iqbal (second author).

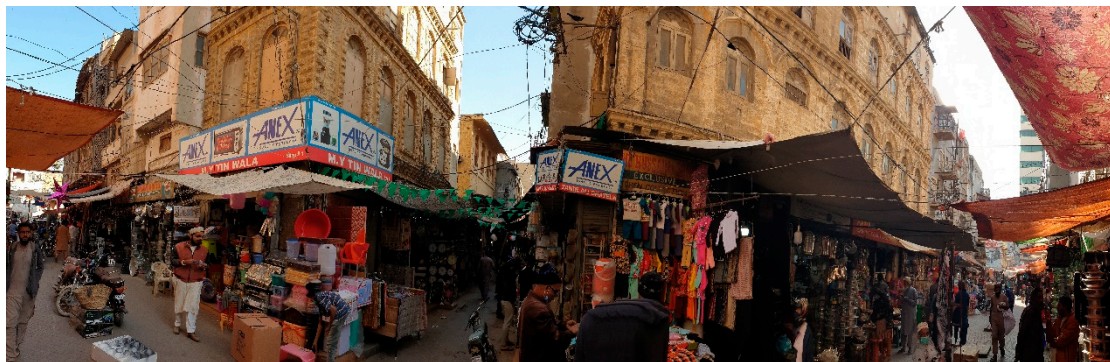

**Figure 18.** Panoramic street view of Bohra Street in Bohri Bazaar showing the condition of British-era buildings and multi-functional whole-sale activities going on in the area. Source: images taken by Syed Hamid Akbar (first author) during 2020–2021 fieldwork.

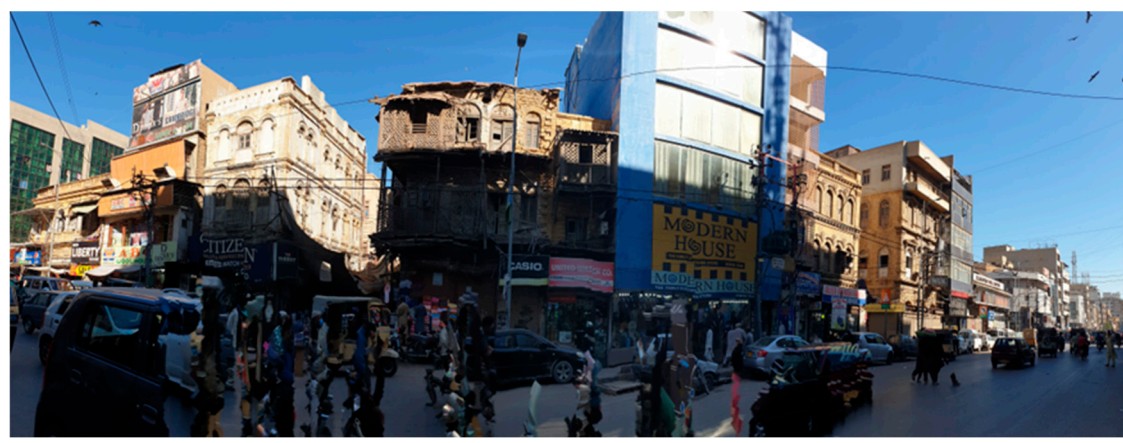

**Figure 19.** Panoramic street view of Zaib-un-Nisa Street showing the condition of British-era buildings and multi-functional commercial activities going on in the area. Source: images taken by Syed Hamid Akbar (first author) during 2020–2021 fieldwork.

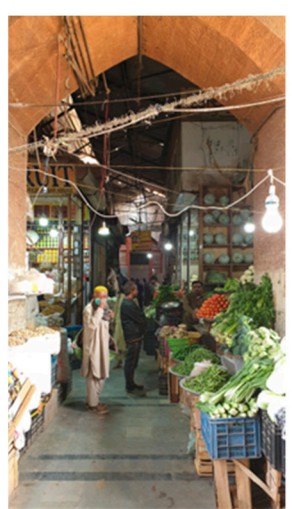 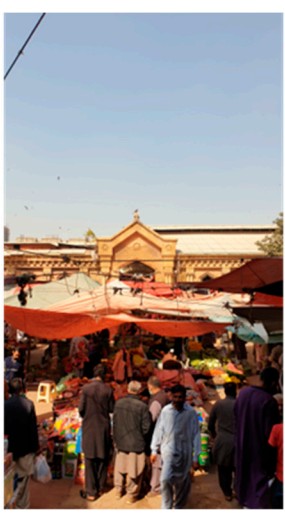 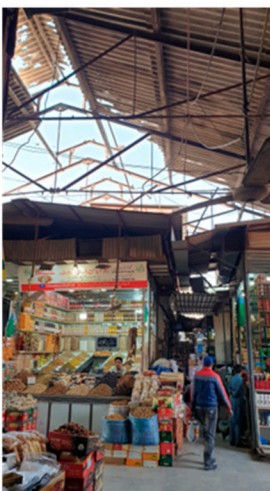

**Figure 20.** Inside pictures of different shops/stalls in Empress Market. Source: images taken by Syed Hamid Akbar (first author) during 2020–2021 fieldwork.

The users of the area and building owners have some emotional, personal connection with the area, and are willing to become part of the process to save these gems. Based on the informal interactions and interviews the first author carried out with the building users,

they are open to complying with the heritage laws. Nevertheless, they need some incentives and guidelines from the authorities to help them survive and boost their business. Many authors, such as [14,15,43], have also pointed out these flaws and the prospect of developing some conservation strategies and management plans to safeguard the British legacy of the Saddar Bazaar Quarter. These authors have also mentioned the lack of heritage laws or the loopholes in the laws and their implementation, which the land developers have used for their benefit and thus were able to demolish many British heritage buildings in Karachi. Despite the heritage laws and the under-process or under-developed heritage-management plan, the buildings' economic values and users association with these buildings, their changing social and cultural values and the increasing land value and pressure from the developers underpin the heritage-management plans. Our findings show that, despite these flaws, the area's British-period layout and buildings can be preserved. Social, cultural, economic and community values can be re-utilised for developing conservation strategies and developing a positive heritage-management plan for the area.

The contemporary adaptive-reuse strategy can be a valuable tool for these buildings. This is not a new concept for the SBQ, as the study shows the informal-reuse aspect of the remaining buildings. The users have adopted intentionally to modify their shops and houses, according to their changing urban, social, cultural or economic requirements. Therefore, this informal-reuse aspect of the building and area can become the inception point and enhance this aspect through the concept of adaptive reuse. From the map in Figure 17, we can see that most high-degree-threat buildings are either attached or nearby each other, in the same street. Therefore, instead of preserving buildings individually, these structures can be presumed to be one continuous historic façade, and concepts such as a living streetscape and a coherent streetscape can be useful as a solution. In addition, through the people-centric or living-heritage approach, the shop and building owners and users of the SBQ can be included in saving these buildings. These strategies can be beneficial to revive Saddar Bazaar's historic urban landscape. In addition, the most affected buildings of the commercial and commercial–residential typologies in the SBQ can be saved from declining conditions towards demolition, and play an important part in sustainable urban development.

*4.3. Research Limitations*

The literature survey, research fieldwork, and visiting different archives were the critical components of our study, and guided the whole process. Over the entire process, accessing the archives and finding the concerned data was not easy. Specifically, the condition of the archival maps made reading and understanding the maps very difficult. In addition, in the archives departments visited in Karachi, some maps and drawings were missing, for which the British Library in London was accessed (online and visited in March 2022). Some British-period buildings that are in use by the military, we were denied access, and not allowed to take photographs for documentation during the field survey.

In some cases, building owners or users did not allow us to take photographs or share any information, and their privacy was respected. For informally interviewing building users, usually, for each building, we tried to interview at least three peeople. Some users, however, refused to interact with us. In addition, some buildings' physical conditions restricted us from visiting the upper floors. These restrictions created some limitations in our research, but did not affect the whole process and outcome of the study.

In conclusion, the map of the SBQ shown in Figure 17 shows the present condition of 76 historic buildings, which were given protection in 1995–1997. Many historic British-period buildings have been demolished, and many are under a high degree of threat. These buildings are at risk of being demolished, thus warning of the loss of an important historical footprint of Karachi's colonial past. This threatens an urban ensemble of considerable quality. It is also understood that a process of 'restoration' is no valid option. Integration of the current informal use—mostly retail and housing—into these historic sites seems the most pressing challenge. Yet, paradoxically, as several of these sites are listed as a

monument, substantial adaptations are not possible. Our research has also observed that legal protection is not necessarily a guarantee of preservation. Finding a new balance between the original monumental appearance and its current, very informal use, is at hand. For this, there is a need for more research on the area's current social and cultural urban layout. Some research on adaptive-reuse concepts and case study projects may also induce a much-needed vocabulary that can be applied to the Saddar Bazaar Quarter.

**Author Contributions:** Conceptual framework, S.H.A.; methodology, S.H.A. and N.I.; research field, visits, S.H.A.; interviews, S.H.A.; literature review S.H.A., N.I.; literature review of SBQ inventory listing, S.H.A.; archival data collection and data evaluation, S.H.A.; photographs, S.H.A.; graphics and map editing, S.H.A. and N.I.; supervision of research, K.V.C.; writing and revising the paper, S.H.A. and N.I.; draft reading and recommendations, K.V.C.; results and discussion, S.H.A. All authors have read and agreed to the published version of the manuscript.

**Funding:** The authors acknowledge the Higher Education Commission (HEC) of Pakistan for providing the necessary funding for the completion of the Ph.D. studies of the first and second authors under the HRDI-UESTP Program, Batch V, Ref: 50035799 and Ref 50035802.

**Data Availability Statement:** Not applicable.

**Acknowledgments:** Special thanks to the support staff of all the archival departments visited for this study, and also to Heritage Cell-DAPNED for providing the authors with the 2006–09 survey reports and plan of the SBQ.

**Conflicts of Interest:** The authors declare no conflict of interest. The funders had no role in the design of the study; in the collection, analyses, or interpretation of data; in the writing of the manuscript; or in the decision to publish the results.

## Notes

1. Official list number from 1995 inventory.
2. 2006–2009 Karachi Heritage Buildings Re-survey Project, a series of unpublished reports produced as separate volumes for each historic quarter, Karachi: Heritage Cell—Department of Architecture and Planning, N.E.D. University. Saddar Bazaar Quarter report was published in February 2008 [29].
3. Good State of Condition identifies well-maintained properties having a homogenous outer appearance, with no alterations that damage or deface the external facades [28].
4. High degree of threat includes buildings that are fifty percent or more vacant, and/ or have 'partially demolished', 'facade only' or 'highly deteriorated' physical condition; thus, they are identified as requiring urgent attention and immediate preventive measures [28].
5. Second-degree threat includes partially maintained properties, those existing in a livable condition, but having gone through changes affecting their external appearance, either due to haphazard alterations or lack of regular maintance and repair work [28].
5. Second-degree threat includes partially maintained properties, those existing in a livable condition, but having gone through changes affecting their external appearance, either due to haphazard alterations or lack of regular maintance and repair work [28].
6. Official list number from 1997 inventory.

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
