# Peer review of "Saddar Bazar Quarter in Karachi: A Case of British-Era Protected Heritage Based on the Literature Review and Fieldwork"

_heritage, doi:10.3390/heritage6030169_

Round 1
Reviewer 1 Report
The paper on Saddar Bazar Quarter in Karachi fits in with the aims and scope of the journal and one of the key topics of this special issue: Institutional governance for heritage management. The article is interesting, and this case has not yet been exhaustively researched, so publications on this topic are valuable. Concluding what policies should be adopted to protect Karachi's heritage is important. On the other hand, the article lacks a complete state of research and a comparison of the results of the presented research with other publications. For this reason, I recommend a major review.
Full comments below:
[Abstract:]
1. In Line 24, the authors suddenly start using the future tense. This gives the impression as if the research has not yet been done.
2. In the abstract, the authors announce future strategies for the site - they did not develop this thread sufficiently in the text.
[Introduction: ]
3. In lines 42-46 please give references to the data and information contained therein.
4. In the following text how many interviews do the authors write about and what were the questions? “The informal interviews were 125 also conducted with the building users and owners with a semi-structured open-ended 126 questionnaire”
[Section 2:]
5. Specific quantitative data (e.g., population) requires providing sources. Not in all places did the authors put them. They require completion. Also present in the paragraph between lines 327-345 is other information for which readers less familiar with the subject would like to be provided with references. I'm thinking here of Pakistani and British law acts.
- I see a general problem with the small number of references given for the condensed information resulting from the literature research. Without the support of specific data, some sentences are unclear or give the impression of arbitrary remarks. I paste below an excerpt from the manuscript with the remarks written in capital letters:
“In response to these unaccepted alterations, some architects, activists, and some pri-365 vate organizations started to raise their voices to protect British-era buildings in Karachi REFERENCE PLEASE. 366 The outcome of their struggle was that a private organization was able to preserve the 367 historic Flagstaff House (now known as Quaid-e-Azam House Museum) WHAT BUILDING IS IT? and landscape WHAT DOES IT MEAN?, 368 and the local authority initiated a list of 44 important buildings under the 'Karachi Build-369 ing REFERENCE PLEASE Control Authority's act of 1979, to give them protection. However, it was in 1994 when 370 a separate Act with the title 'Sindh Cultural Heritage Preservation Act of 1994' was passed REFERENCE TO THE ACT. 371 And under the act, initial inventories were made from 1995-1997, listing 581 historic 372 buildings from nineteen historic quarters of Karachi that were given protected heritage 373 status. [14]–[16]. The Saddar Bazaar quarter was also among the nineteen quarters, and 76 374 buildings of British time were included in the enlistment for protection based on their 375 historical values REFERENCE PLEASE. The buildings were from different typologies such as residential, com-376 mercial, residential cum commercial, or commercial cum residential buildings. This was 377 a new concept for Pakistan, specifically in Karachi, where till now, only a few monumental 378 masterpieces were considered buildings with historical values REFERENCE PLEASE. (Table 4).”
7. In table 4 putting the row with a type and number of buildings – “monumental masterpieces” – would be good to compare the authors’ calculations.
- Do you have any links to these survey results?: “A survey conducted by Heritage cell at NED University Karachi 403 from 2006 to 2009 documented and reported that of the 76 buildings which were given 404 protected heritage status through the enlistment of 1995-97 in SBQ, only fourteen per cent 405 were in good state of condition, and at the same time thirty-six per cent was facing high 406 degree threatii. (Figure 10).
[Section 3:]
9. The word “discussion” appears in the title of section 3. It looks like the real discussion is in section 4.
- I would say that the following text from Section 3.1 belongs to the “method” section: “To find the present condition of these build-426 ings and investigate the reasons behind their current state, research visits were done in 427 January 2019 and again from December 2020 to January 2021. During the fieldwork of 428 2020-2021, Saddar Bazaar quarter was visited several times, and photographs were taken 429 to document the present condition of 76 buildings from 1995-1997 enlistment. These pic-430 tures were also compared to the previous reports and the historic photos from different 431 archives.”
- Table 5 – there is an error with one date.
- What are the criteria for assigning a building to Second Degree or High Degree Threat? Where are they derived from?
[Discussion:]
13. Map (figure 17) and its analysis in the text belongs rather to the results that the discussion. On the other hand, the discussion should include comparisons of the authors' findings with previously published studies on this case or similar ones. This brings us to published studies that the authors did not mention. The following should be mentioned here: T. A. Soomro, M. A. Soomro, A. N. Laghari, D. K. Bhangwar, and M. A. Soomro, “Fading Legacy of the Architectural Heritage of the Historic Core of Karachi”, Eng. Technol. Appl. Sci. Res., vol. 8, no. 2, pp. 2735–2740, Apr. 2018, “The Lost Camp OF Karachi: A Case study of Saddar Bazaar- The Earliest British Settlement” by Hira Ovais (https://www.researchgate.net/publication/366464178_The_Lost_Camp_OF_Karachi_A_Casestudy_of_Saddar_Bazaar-The_Earliest_British_Settlement), or “THE HAWKERS OF SADDAR BAZAAR. A Plan for the Revitalisation of Saddar Bazaar Karachi Through Traffic Rerouting and the Rehabilitation of its Hawkers” by Arif Hasan, Asiya Sadiq Polak I Christophe Polak.
- The discussion should also include a limitation of the research.
[Figures and tables:]
15. The order of the illustrations is mixed up in illustrations 2 and 3. Illustration 3 is small and thus unreadable. The same applies to the other illustrations as well.
16. Figure 8: Is it possible to enlarge the photographs so that readers can see more clearly the changes that the authors write about in the text?
Author Response
[Abstract:]
- In Line 24, the authors suddenly start using the future tense. This gives the impression as if the research has not yet been done.
Response: Corrected. Line 26
- In the abstract, the authors announce future strategies for the site - they did not develop this thread sufficiently in the text.
Response: We wanted to discuss some strategies in general, as this paper is part of a broder Ph.D. research. So based on outcomes of this paper we are working on proposing some strategies in details as next step. So, we have rephrased sentence. Lines 30-33
[Introduction: ]
- In lines 42-46 please give references to the data and information contained therein.
Response:References added, and text corrected accordingly. Lines 46-51.
- In the following text how many interviews do the authors write about and what were the questions? “The informal interviews were 125 also conducted with the building users and owners with a semi-structured open-ended 126 questionnaire”
Response:We interviewed the users and building owners of the structures which are still remaining from the 1995-97 heritage inventory list. Usually for each building we tried to interview at least three persons. But due to limitation of research it was not possible for some buildings. We have mentioned that in the newly added research limitation section. Also questions which were asked but not limited to has been included. Lines 142-160, and also Section 4.3. Lines 746-748
[Section 2:]
- Specific quantitative data (e.g., population) requires providing sources. Not in all places did the authors put them. They require completion. Also present in the paragraph between lines 327-345 is other information for which readers less familiar with the subject would like to be provided with references. I'm thinking here of Pakistani and British law acts.
Response:All the sources have been added with the quantitative data( population) where required.
More references has been added to clarify the information between lines 327-345. Check lines 381-403.
- I see a general problem with the small number of references given for the condensed information resulting from the literature research. Without the support of specific data, some sentences are unclear or give the impression of arbitrary remarks. I paste below an excerpt from the manuscript with the remarks written in capital letters:
“In response to these unaccepted alterations, some architects, activists, and some pri-365 vate organizations started to raise their voices to protect British-era buildings in Karachi REFERENCE PLEASE. 366 The outcome of their struggle was that a private organization was able to preserve the 367 historic Flagstaff House (now known as Quaid-e-Azam House Museum) WHAT BUILDING IS IT? and landscape WHAT DOES IT MEAN?, 368 and the local authority initiated a list of 44 important buildings under the 'Karachi Build-369 ing REFERENCE PLEASE Control Authority's act of 1979, to give them protection. However, it was in 1994 when 370 a separate Act with the title 'Sindh Cultural Heritage Preservation Act of 1994' was passed REFERENCE TO THE ACT. 371 And under the act, initial inventories were made from 1995-1997, listing 581 historic 372 buildings from nineteen historic quarters of Karachi that were given protected heritage 373 status. [14]–[16]. The Saddar Bazaar quarter was also among the nineteen quarters, and 76 374 buildings of British time were included in the enlistment for protection based on their 375 historical values REFERENCE PLEASE. The buildings were from different typologies such as residential, com-376 mercial, residential cum commercial, or commercial cum residential buildings. This was 377 a new concept for Pakistan, specifically in Karachi, where till now, only a few monumental 378 masterpieces were considered buildings with historical values REFERENCE PLEASE. (Table 4).”
Response:References has been added to support the information, and paragraph is been corrected as per changes. Lines 425-449
- In table 4 putting the row with a type and number of buildings – “monumental masterpieces” – would be good to compare the authors’ calculations.
Response:The table for is based upon a previous report/survey done by Heritage Cell-DAPNED in 2008. The report does not separately mention the number off monumental masterpieces which were given protected status. So, instead of adding a row we have put the name of the buildings which are considered master pieces (based on their architecture, scale and function). See lines 446-447.
- Do you have any links to these survey results?: “A survey conducted by Heritage cell at NED University Karachi 403 from 2006 to 2009 documented and reported that of the 76 buildings which were given 404 protected heritage status through the enlistment of 1995-97 in SBQ, only fourteen per cent 405 were in good state of condition, and at the same time thirty-six per cent was facing high 406 degree threatii. (Figure 10).
Response:Yes, we have added the reference of these survey in the caption of the Table-4 and figure-10. Also endnote is added to further support it. See line 477 and endnote ii.
[Section 3:]
- The word “discussion” appears in the title of section 3. It looks like the real discussion is in section 4.
Response: We corrected the title. See line 485.
- I would say that the following text from Section 3.1 belongs to the “method” section: “To find the present condition of these build-426 ings and investigate the reasons behind their current state, research visits were done in 427 January 2019 and again from December 2020 to January 2021. During the fieldwork of 428 2020-2021, Saddar Bazaar quarter was visited several times, and photographs were taken 429 to document the present condition of 76 buildings from 1995-1997 enlistment. These pic-430 tures were also compared to the previous reports and the historic photos from different 431 archives.”
Response: We have mention that in the methodology section. Some information was a repetition, so we deleted that that and moved some to method section. See lines 141-142, and 499-505.
- Table 5 – there is an error with one date. Response: Corrected date error in table-5.
- What are the criteria for assigning a building to Second Degree or High Degree Threat? Where are they derived from?
Response: The criteria for different degree of threat are derived from a previous study. We have added that through endnotes and also put the references to it . See endnotes iii, iv and v. Also reference (28).
[Discussion:]
- Map (figure 17) and its analysis in the text belongs rather to the results that the discussion. On the other hand, the discussion should include comparisons of the authors' findings with previously published studies on this case or similar ones. This brings us to published studies that the authors did not mention. The following should be mentioned here: T. A. Soomro, M. A. Soomro, A. N. Laghari, D. K. Bhangwar, and M. A. Soomro, “Fading Legacy of the Architectural Heritage of the Historic Core of Karachi”, Eng. Technol. Appl. Sci. Res., vol. 8, no. 2, pp. 2735–2740, Apr. 2018, “The Lost Camp OF Karachi: A Case study of Saddar Bazaar- The Earliest British Settlement” by Hira Ovais
(https://www.researchgate.net/publication/366464178_The_Lost_Camp_OF_Karachi_A_Casestudy_of_Saddar_Bazaar-The_Earliest_British_Settlement), or “THE HAWKERS OF SADDAR BAZAAR. A Plan for the Revitalisation of Saddar Bazaar Karachi Through Traffic Rerouting and the Rehabilitation of its Hawkers” by Arif Hasan, Asiya Sadiq Polak I Christophe Polak.
Response:We have rephrased, improved the Section 4. It is been divided into three sub-sections:
- Results: See lines 650-680.
- Discussion: See lines 681-735.
- Research limitations: See lines 736-750
Thanks for your valuable comments and suggestions of previously published studies. We have compared my findings and results with the two suggested papers. The book you have suggested ‘ The Hawkers of the Saddar Bazaar’ is a proposed traffic plan based on their study. As this paper is part of a broder Ph.D. research. So based on outcomes of this paper we are working on proposing some strategies in details as next step. So, we think this book will be more helped to be compared by our proposed strategies, which is our second paper and work is in progress on it.
- The discussion should also include a limitation of the research.
Response:See 4.3. Research limitations: See lines 736-750 .
[Figures and tables:]
- The order of the illustrations is mixed up in illustrations 2 and 3. Illustration 3 is small and thus unreadable. The same applies to the other illustrations as well.
Response:The order of illustrations is corrected. And all images size are increased .
- Figure 8: Is it possible to enlarge the photographs so that readers can see more clearly the changes that the authors write about in the text?
Response: And all images size are increased .

Reviewer 2 Report
heritage-2211391-peer-review-v1
Review of “
Saddar Bazar Quarter in Karachi : A case of British era Protected Heritage based on the literature review and Fieldwork”
Overall this is very valuable contribution to the understanding of heritage management issues in Pakistan and thus has much merit. One major aspect that is missing in the paper is a more nuanced discussion of the community and heritage values that underpin heritage management.
Given that personal community values underpin all of this, from the election of city officials to the community support of city management actions, it is important to further elaborate on the projected nature of heritage values (https://doi.org/10.1179/2159032X13Z.00000000011 (https://doi.org/10.1080/13527258.2011.619554 ) and as the fact these are changing intergenerationally (see https://www.mdpi.com/2571-9408/5/3/105 ).
Also missed is an opportunity to discuss in more depth whether the fact that these are British, colonial era structures, rather than post-Independence structures plays a role here
Line 406 what is a good state of preservation be more specific in your definition of the classification of the threat categories
The images of the historic buildings in figures 5 and 6 should be paired with modern day (2022/2023) images as was done in figure 9
Ethics
As the authors formally interviewed several architects for the study, the authors noted to provide evidence that ethics approval had been sought and obtained from the Ethics Review Board of their academic institution, or evidence from the Ethics Review Board that that approval is not required for the study as designed. In this day and age this approval must be referenced /stated in the paper.
Referencing
The bibliographic referencing of figure 3 is inadequate, needs formal publication details
The bibliographic referencing of figure 2 is inadequate, need library item number
The bibliographic referencing of figure 4 is inadequate, needs formal publication details
The bibliographic referencing of figure 5 is inadequate, need library item numbers
The bibliographic referencing of figure 6 is inadequate, need library item numbers
Minor issues
Figure 2 is out of sequence
Line 14 expression: “ mud-fortified small town”
Line 42 [4, p.].
Line 49 stray full stop: demolition. [1]
Line 61 expression: “became the Sindh province's capital” became capital of the Sindh province
Line 148 expression: “Karachi till 1839” Karachi until 1839
Line 244 “European” should be European
Line 365 expression: “In response to these unaccepted alterations” unacceptable?
Author Response
- Overall this is very valuable contribution to the understanding of heritage management issues in Pakistan and thus has much merit. One major aspect that is missing in the paper is a more nuanced discussion of the community and heritage values that underpin heritage management. Given that personal community values underpin all of this, from the election of city officials to the community support of city management actions, it is important to further elaborate on the projected nature of heritage values (https://doi.org/10.1179/2159032X13Z.00000000011 (https://doi.org/10.1080/13527258.2011.619554 ) and as the fact these are changing intergenerationally (see https://www.mdpi.com/2571-9408/5/3/105 ).
Response: Very useful comment regarding adding the discussion on community and heritage values impact on heritage management. As this paper is part of a broder Ph.D., and as our next step based on the results of this paper we are working to develop a separate paper through which we want propose a vocabulary that can be adopted in Saddar Bazaar. We have also mentioned that in conclusion (line737-748). But your comment also helped us to thinking of adding a little discussion in this paper so we have added and improved our discussion section. See lines 591-599 and also 4.2. Discussion. See lines 681-735.
- Also missed is an opportunity to discuss in more depth whether the fact that these are British, colonial era structures, rather than post-Independence structures plays a role here.
Response: See the improved section 4.2. Discussion. See lines 681-735. Figure 18-20 also added to support the text.
- Line 406 what is a good state of preservation be more specific in your definition of the classification of the threat categories
Response: The criteria for different degree of threat are derived from a previous study. We have added that through endnotes and also put the references to it . See endnotes iii, iv and v. Also reference (28).
- The images of the historic buildings in figures 5 and 6 should be paired with modern day (2022/2023) images as was done in figure 9
Response:For figure 5, due to some limitation of research we were not able to visit and photograph those sites during fieldwork of 2020/21. See lines 742-744.
For figure 6 we have done.
- Ethics
As the authors formally interviewed several architects for the study, the authors noted to provide evidence that ethics approval had been sought and obtained from the Ethics Review Board of their academic institution, or evidence from the Ethics Review Board that that approval is not required for the study as designed. In this day and age this approval must be referenced /stated in the paper.
Response: The methodology which I used to interact with the people is based on informal interviews and open ended questions. I did not make structured questionnaires to share with the participants. Our research did not target specific people. I went to visit different departments and universities to collect data, and during that I met some people there so we started a discussion on my research and during that discussion I asked informal open ended questions based on their responses. So, it was an informal interaction.
Before going on my fieldwork to Pakistan, I explained this process to my supervisor and other committee members. This was the first step of my doctoral research and the purpose of this informal interaction was to get an idea of whether the proposed topic has the potential to be worked on, and then finalize my first research proposal to be submitted for the approval of the faculty council. So, they were not planned interviews, and I was also not sure with whom I will be able to meet. The interviewees were met randomly, and a general discussion and open ended questions was done with them. We will also not use the names of the persons , just their profession and the date when we met them. See Lines 114-117, and table-1.
I hope this explanation will be enough.
- Referencing
- The bibliographic referencing of figure 3 is inadequate, needs formal publication details. Response: Details added.
- The bibliographic referencing of figure 2 is inadequate, need library item number. Response: Downloaded from British library Online Gallery : Album name is added.
- The bibliographic referencing of figure 4 is inadequate, needs formal publication details. Response:Details added.
- The bibliographic referencing of figure 5 is inadequate, need library item numbers. Response:Downloaded from British library Online Gallery : Album name is added.
- The bibliographic referencing of figure 6 is inadequate, need library item numbers. Response:Downloaded from British library Online Gallery : Album name is added.
- Minor issues
- Figure 2 is out of sequence. Response: Corrected
- Line 14 expression: “ mud-fortified small town”. Response: This is the expression which many historians and writers have used in previously publications.
- Line 42 [4, p.]. Response: See line 46.
- Line 49 stray full stop: demolition. [1] Response: See line 54.
- Line 61 expression: “became the Sindh province's capital” became capital of the Sindh province Response: See line 68.
- Line 148 expression: “Karachi till 1839” Karachi until 1839. Response: See line 182
- Line 244 “European” should be European. Response: See line 282
- Line 365 expression: “In response to these unaccepted alterations” unacceptable? Response: See line 425.

Round 2
Reviewer 1 Report
Thank you for the corrections suggested in the review.
Author Response
Comments and Suggestions for Authors:Thank you for the corrections suggested in the review. Response: Respected reviewer, thanks for you wise comments that improved my manuscript.
Reviewer 2 Report
The authors have adequately addressed most of my concerns. The paper can be published in this version
Author Response
Comments and Suggestions for Authors: The authors have adequately addressed most of my concerns. The paper can be published in this versionResponse: Respected reviewer, thanks for you wise comments and for suggesting paper for publication. Your comments improved my manuscript.